# Do We Need to Verify Step by Step?
# Rethinking Process Supervision from a Theoretical Perspective

**Zeyu Jia** [1]  **Alexander Rakhlin** [1]  **Tengyang Xie** [2]

## Abstract

As large language models have evolved, it has become crucial to distinguish between process supervision and outcome supervision—two key reinforcement learning approaches to complex reasoning tasks. While process supervision offers intuitive advantages for long-term credit assignment, the precise relationship between these paradigms has remained an open question. Conventional wisdom suggests that outcome supervision is fundamentally more challenging due to the trajectory-level coverage problem, leading to significant investment in collecting fine-grained process supervision data. In this paper, we take steps towards resolving this debate. Our main theorem shows that, under standard data coverage assumptions, *reinforcement learning through outcome supervision is no more statistically difficult than through process supervision*, up to polynomial factors in horizon. At the core of this result lies the novel *Change of Trajectory Measure Lemma*—a technical tool that bridges return-based trajectory measure and step-level distribution shift. Furthermore, for settings with access to a verifier or a rollout capability, we prove that any policy's advantage function can serve as an optimal process reward model, providing a direct connection between outcome and process supervision. These findings suggest that the empirically observed performance gap—if any—between outcome and process supervision likely stems from algorithmic limitations rather than inherent statistical difficulties, potentially transforming how we approach data and algorithm design for reinforcement learning.

[1]Massachusetts Institute of Technology, Cambridge MA, USA [2]University of Wisconsin–Madison, Madison WI, USA. Correspondence to: Zeyu Jia <zyjia@mit.edu>, Alexander Rakhlin <rakhlin@mit.edu>, Tengyang Xie <tx@cs.wisc.edu>.

*Proceedings of the $42^{nd}$ International Conference on Machine Learning*, Vancouver, Canada. PMLR 267, 2025. Copyright 2025 by the author(s).

## 1. Introduction

Reward signals play a central role in reinforcement learning, and it has been hypothesized that intelligence and its associated abilities could emerge naturally from the simple principle of reward maximization (Silver et al., 2021). Over the past decade, this idea has been powerfully demonstrated across diverse AI systems. In specialized domains like AlphaGo Zero (Silver et al., 2017), superhuman performance has been achieved by maximizing simple, well-defined environmental reward signals. The paradigm has also proven transformative for general-purpose AI systems, particularly in training large language models (LLMs) using reinforcement learning (Ouyang et al., 2022; Bai et al., 2022; Jaech et al., 2024). However, for these more open-ended systems, the challenge of reward specification is significantly more complex, requiring reward signals to be learned from human-annotated data through reward modeling rather than being manually specified.

This challenge of reward specification has led to the emergence of two fundamental supervision paradigms in reinforcement learning (e.g., Uesato et al., 2022; Lightman et al., 2023):

- *Outcome supervision:* Reward feedback is provided only after the final output, based on the final outcomes or—in the case of LLMs—overall quality of the model's chain-of-thought (CoT).

- *Process supervision:* Fine-grained reward feedback is provided based on the quality of each intermediate step (e.g., correctness of each step in the CoT in the case of LLMs).

The choice between these paradigms represents a fundamental trade-off in reinforcement learning system design. Process supervision offers several intuitive advantages: it provides more granular feedback, enables better interpretability of model decisions, and potentially allows for more efficient credit assignment in long reasoning chains. These benefits have led to significant investment in collecting step-by-step feedback data, despite the substantial human effort required (Lightman et al., 2023). The granular nature of process supervision also aligns with how humans often learn and teach—through step-by-step guidance rather than just final outcomes.

However, the high cost of collecting process supervision data raises important questions about its necessity. Outcome supervision, while providing less detailed feedback, offers practical advantages in terms of data collection efficiency and scalability. It also reflects various natural settings of human learning where detailed step-by-step feedback may not be available (e.g., game-playing). Recent empirical advances in automated process supervision derived from outcome supervision (Wang et al., 2024; Luo et al., 2024; Zhong et al., 2024; Yuan et al., 2024; Setlur et al., 2024) or in directly learning from outcome supervision (Guo et al., 2025) suggest that the statistical benefits of process supervision might not be as fundamental as previously thought.

This tension between process and outcome supervision touches on deeper questions in machine learning and cognitive science: How much explicit supervision is truly necessary for effective learning? Can systems learn optimal behavior from sparse reward signals, or is step-by-step guidance fundamentally necessary? Understanding these questions has important implications not just for practical system design, but also for our broader understanding of learning and intelligence.

### 1.1. Our Results

This paper examines the statistical performance of reinforcement learning under outcome supervision—an emerging paradigm that has garnered significant attention in large language model research. Our findings challenge conventional wisdom that outcome supervision is inherently more difficult than process supervision due to its coarser feedback.

(1) Our main results (Section 3) demonstrate that given a dataset of trajectories with only cumulative rewards (as in outcome supervision), we can transform the data into trajectories with per-step rewards, while only paying an additive error that scales with the *state-action concentrability*. As a result, we can transform any algorithm that takes trajectories with per-step rewards as input into an algorithm that takes trajectories with total rewards as input, with essentially no loss in statistical performance up to polynomial factors of the horizon.

(2) We also provide (Section 4) a theoretical analysis of using Q-functions or advantage functions as reward functions, a popular approach in practice for mimicking process supervision from outcome supervision data (e.g., Wang et al., 2024; Luo et al., 2024; Setlur et al., 2024). We prove that the advantage function of *any policy* can serve as an optimal process reward model; in contradistinction, using Q-functions could lead to sub-optimal results.

Beyond these two main messages, we make the following contributions:

(1) Our key technical contribution—*Change of Trajectory Measure Lemma* (Lemma 3)—is applicable beyond our main results. The change of measure is a fundamental operation in analyzing off-policy evaluation (e.g., Uehara et al., 2020), offline reinforcement learning (e.g., Xie et al., 2021a), and out-of-domain generalization (Dong and Ma, 2022). To our knowledge, Lemma 3 presents the first result concerning trajectory-level change of measure via step-level distribution shift.

(2) We also extend our main results to the setting of preference-based reinforcement learning (Section 3.4). Namely, we transform preference-based trajectory data, generated according to the Bradley-Terry model, into a dataset of trajectories with per-step reward. In particular, for direct preference optimization (Rafailov et al., 2023), we improve the previous analyses and show that its sample complexity only scales with the state-action concentrability coefficients instead of trajectory concentrability coefficients—potentially, an exponential improvement.

### 1.2. Notation

We use $a_n \lesssim b_n$ or $a_n = \mathcal{O}(b_n)$ whenever there exists some universal positive constant $c$ such that $a_n \leq c \cdot b_n$. We use $\sigma : \mathbb{R} \to [0, 1]$ to denote the sigmoid function $x \mapsto \sigma(x) = \exp(x)/(1 + \exp(x))$.

## 2. Background

In this section, we introduce key prerequisite concepts. We begin with the basics of Markov Decision Processes (Section 2.1). We then discuss the two aforementioned supervision paradigms in reinforcement learning (Section 2.2). Finally, we review the concepts of state-action and trajectory concentrability (Section 2.3).

### 2.1. Markov Decision Processes

An MDP $M$ consists of a tuple $(\mathcal{S}, \mathcal{A}, P, r^\star, H)$. Here $H \in \mathbb{Z}_+$ denotes the horizon, $\mathcal{S} = \cup_{h=1}^H \mathcal{S}_h$ denotes the layered state space, $\mathcal{A}$ denotes the action space, $P = (P_1, \cdots, P_H)$ with $P_h : \mathcal{S}_h \times \mathcal{A} \to \Delta(\mathcal{S}_{h+1})$ denoting the transition model, and $r^\star : \mathcal{S} \times \mathcal{A} \to [0, 1]$ is the deterministic ground truth reward model. For simplicity, we let $s_1 \in \mathcal{S}_1$ be a fixed initial state.

For a trajectory $\tau = (s_1, a_1, s_2, a_2, \cdots, s_H, a_H)$, we write $r(\tau) := \sum_{h=1}^H r(s_h, a_h)$ to denote the total reward accumulated along the trajectory under deterministic reward model $r : \mathcal{S} \times \mathcal{A} \to \mathbb{R}$ (which can be the ground truth reward model $r^\star$ or any learned reward model $\hat{r}$). A (Markov) policy $\pi : \mathcal{S} \to \Delta(\mathcal{A})$ is a mapping from the state space $\mathcal{S}$ to a distribution on the action space $\mathcal{A}$. The notation $\mathbb{P}_{\tau \sim \pi}$ and $\mathbb{E}_{\tau \sim \pi}$ stands for the probability and expectation

with respect to trajectories $\tau$ sampled according to policy $\pi$ within the transition model given by $P$, starting from a fixed state $s_1$. For any given policy $\pi$, the occupancy measure of any state $s_h \in \mathcal{S}_h$ in layer $h$ and any state-action pair $(s_h, a_h) \in \mathcal{S}_h \times \mathcal{A}$ are defined, respectively, as $d^\pi(s_h) := \mathbb{P}_{\tau \sim \pi}(s_h \in \tau)$ and $d^\pi(s_h, a_h) := \mathbb{P}_{\tau \sim \pi}((s_h, a_h) \in \tau)$. Additionally, we define the trajectory occupancy measure for any trajectory $\tau$, $d^\pi(\tau) := \mathbb{P}_{\tau' \sim \pi}(\tau = \tau')$, as well as $\pi(\tau) := \prod_{(s_h, a_h) \in \tau} \pi(a_h \mid s_h)$ (note that $d^\pi(\tau)$ and $\pi(\tau)$ are different when transitions is stochastic). For any policy $\pi$, we use $J(\pi)$ to denote the expected total reward of trajectories collected by $\pi$ under the ground truth reward $r^\star$, i.e., $J(\pi) := \mathbb{E}_{\tau \sim \pi}[r^\star(\tau)]$. For a specific reward model $r$, we use $J_r(\pi)$ to denote the expected total reward of trajectories collected by $\pi$ under reward $r$, i.e., $J_r(\pi) := \mathbb{E}_{\tau \sim \pi}[r(\tau)]$. We assume that $r^\star(\tau) \in [0, 1]$ for any trajectory $\tau$.

## 2.2. Outcome Supervision and Process Supervision

This paper focuses on two basic supervision paradigms in reinforcement learning: *process supervision* and *outcome supervision*. We analyze these approaches through the lens of *statistical complexity* rather than algorithmic implementation details.

The distinction lies in the temporal resolution of available reward signals:

- Process supervision provides step-wise rewards during trajectory collection. More precisely, the offline data has the form

$$\mathcal{D}_P := \{(s_1, a_1, r_1, s_2, a_2, r_2, \cdots, s_H, a_H, r_H)\}, \quad (1)$$

  where $r_h = r^\star(s_h, a_h)$ denotes the ground truth reward value at step $h$. This setting is compelling compared to outcome supervision, especially for complex multi-step reasoning problems, as it provides more precise feedback that can pinpoint the location of suboptimal actions and allows to correct for cases where the agent makes mistakes in the middle of the reasoning path but reaches the correct final answer (Uesato et al., 2022; Lightman et al., 2023).

- Outcome supervision reveals only the cumulative rewards for complete trajectories. The data for this setting has the form

$$\mathcal{D}_O := \{(s_1, a_1, s_2, a_2, \cdots, s_H, a_H, R)\}, \quad (2)$$

  where $R = \sum_{h=1}^H r^\star(s_h, a_h)$ denotes the total reward along the trajectory $(s_1, a_1, ..., s_H, a_H)$.

We also consider preference learning, where the learner only has access to trajectory-level pairwise preferences, as an extension of outcome supervision. Our main results are applicable to both cases, and are discussed in Section 3. The problem of reinforcement learning from human preferences (e.g., Christiano et al., 2017)

or human feedback (e.g., Ouyang et al., 2022) typically falls under this paradigm, where only the trajectory-level supervision signal is available.

Our analysis reveals the temporal resolution distinction described above does not inherently create statistical complexity gaps when proper coverage conditions hold. Our results formalize this insight in the following two settings: (1) Offline RL with different reward signal resolutions (Section 3), and (2) Online RL with verifier/rollout access (Section 4).

**Outcome-Supervised and Process-Supervised Reward Models.** Two other commonly used terms related to this paper are outcome-supervised reward models (ORMs) and process-supervised reward models (PRMs). ORMs are learned to evaluate the final outcomes of whole trajectories, corresponding to the value $R$ in Eq. (2). PRMs are learned to evaluate the intermediate rewards of each state-action pair, corresponding to the values $r^\star(s_h, a_h)$ in Eq. (1) for all $h \in [H]$.

In this paper, we use ORMs and PRMs to refer specifically to different algorithmic approaches for learning reward models, rather than the underlying supervision paradigms discussed earlier, as learning ORMs or PRMs does not necessarily require the underlying data to be collected under the same supervision paradigm.

## 2.3. State-Action Coverage and Trajectory Coverage

The coverage condition—typically referred to as a bounded *concentrability coefficient* (Munos, 2003; Antos et al., 2008; Farahmand et al., 2010; Chen and Jiang, 2019; Jin et al., 2021; Xie and Jiang, 2021; Xie et al., 2021a; Bhardwaj et al., 2023)—has played a central role in the theory of offline (or, batch) reinforcement learning, and has recently gained growing attention in online reinforcement learning through a related concept of a *coverability coefficient* (Xie et al., 2022b; Liu et al., 2023; Amortila et al., 2024a;b).

In this paper, we use coverage conditions to capture the statistical complexity of different supervision paradigms. To motivate the importance of coverage notions, consider the following approach for imputing the missing rewards in outcome supervision. Suppose we minimize the trajectory-level regression objective over a class of reward functions $\mathcal{R} = \{r : \mathcal{S} \times \mathcal{A} \to \mathbb{R}\}$,

$$\widehat{r} = \arg\min_{r \in \mathcal{R}} \sum_{(\tau, R) \in \mathcal{D}_O} (r(\tau) - R)^2, \quad (3)$$

where the outcome supervision data $\mathcal{D}_O$ are collected by some reference policy $\pi_{\text{off}}$. With the learned reward model $\widehat{r}$, we can now employ an offline RL method of our choice. However, we have a (standard in the literature) mismatch: while $\left| \sum_{h=1}^H \widehat{r}(s_h, a_h) - \sum_{h=1}^H r^\star(s_h, a_h) \right|$ is small for trajectories collected by $\pi_{\text{off}}$, we care about the error $\left| J(\pi) - J_{\widehat{r}}(\pi) \right|$ for some policy $\pi$ that differs from

$\pi_{\text{off}}$, where $J(\pi)$ and $J_{\widehat{r}}(\pi)$ are defined in Section 2.1. A naive approach for capturing such a change of trajectory measure is to use the *trajectory concentrability coefficient*,

$$C_{\text{traj}}(\pi, \pi_{\text{off}}) := \sup_{\tau} \frac{d^{\pi}(\tau)}{d^{\pi_{\text{off}}}(\tau)}, \tag{4}$$

where the supremum is over all possible trajectories.

This trajectory concentrability coefficient is usually considered to be prohibitively large, and its limitation has been widely studied in the literature on off-policy evaluation (e.g., Liu et al., 2018; Xie et al., 2019; Nachum et al., 2019; Uehara et al., 2020). An alterative approach is to express upper bounds (if possible) in terms of the *state-action concentrability coefficient*, commonly used in offline policy learning literature (e.g., Munos, 2003; Antos et al., 2008; Farahmand et al., 2010; Chen and Jiang, 2019) and defined as follows:

$$C_{\text{sa}}(\pi, \pi_{\text{off}}) := \max_{h \in [H]} \sup_{s_h \in \mathcal{S}_h, a \in \mathcal{A}} \frac{d^{\pi}(s_h, a_h)}{d^{\pi_{\text{off}}}(s_h, a_h)}. \tag{5}$$

The state-action concentrability coefficient is always smaller than the trajectory concentrability coefficient, and the difference can be exponential. This is because $d^{\pi}(s_h, a_h)$ aggregates over all trajectories that pass through $(s_h, a_h)$, and thus

$$\frac{d^{\pi}(s_h, a_h)}{d^{\pi_{\text{off}}}(s_h, a_h)} = \frac{\sum_{\tau:(s_h, a_h) \in \tau} d^{\pi}(\tau)}{\sum_{\tau:(s_h, a_h) \in \tau} d^{\pi_{\text{off}}}(\tau)} \le \sup_{\tau} \frac{d^{\pi}(\tau)}{d^{\pi_{\text{off}}}(\tau)},$$

for all $h \in [H]$ and $(s_h, a_h) \in \mathcal{S}_h \times \mathcal{A}$.

It is straightforward to see that using the state-action concentrability coefficient is usually sufficient for the process supervision paradigm, the setting prevalent in the offline RL literature. The intuition is that process supervision allows us to achieve small estimation error at the state-action level, which only poses a state-action level change of measure during the subsequent policy learning step. *Perhaps surprisingly, we show that state-action concentrability is also sufficient for the outcome supervision case.*

**Single-Policy and All-Policy Coverage.** Another important concept in the offline RL literature is the distinction between single-policy and all-policy concentrability. Using the state-action concentrability coefficient as an example, single-policy concentrability refers to the coverage of a specific target policy (such as the optimal policy $\pi^{\star}$), defined as $C_{\text{sa}}(\pi^{\star}, \pi_{\text{off}})$. In contrast, all-policy concentrability considers the coverage of all policies in a policy class $\Pi$, defined as $C_{\text{sa}}(\Pi, \pi_{\text{off}}) := \sup_{\pi \in \Pi} C_{\text{sa}}(\pi, \pi_{\text{off}})$.

Single-policy versus all-policy coverage is one of the central questions in the offline RL literature. For more details, we refer the reader to Chen and Jiang (2019); Xie et al. (2021a); Jiang and Xie (2024), but this is not the focus of this paper.

In the following sections, we will present results using all-policy coverage conditions for simplicity, and defer the single-policy case to the appendix.

## 3. Outcome and Process Supervision: Similar Statistical Guarantees

### 3.1. Learning a Reward Model from Total Reward

We present a simple approach to estimate rewards in an outcome supervision dataset of the form Eq. (2) using least squares regression, assuming that the learner has access to a class of reward models $\mathcal{R}$. This transformation allows the learner to use outcome supervision data with methods designed for process reward data, as detailed below. We have the following theorem for the least squares estimate of the rewards:

**Theorem 1.** *Suppose the dataset $\mathcal{D}_O$ is collected i.i.d. according to policy $\pi_{\text{off}}$ in the MDP $M = (\mathcal{S}, \mathcal{A}, P, r^{\star}, H)$ with the ground truth reward model $r^{\star} \in \mathcal{R}$. Then, with probability at least $1 - \delta$, for any policy $\pi$, the PRM reward model $\widehat{r}$ computed by Eq. (3) satisfies*

$$|J_{\widehat{r}}(\pi) - J(\pi)| \lesssim H^{3/2} \cdot \sqrt{\frac{C_{\text{sa}}(\pi, \pi_{\text{off}}) \cdot \log(|\mathcal{R}|/\delta)}{|\mathcal{D}_O|}},$$

*where $C_{\text{sa}}(\pi, \pi_{\text{off}})$ is the state-action concentrability coefficient defined in Eq. (5).*

The proof of Theorem 1 is deferred to Appendix B.2. Theorem 1 yields an approach for transforming any offline RL algorithm which takes trajectories with per-step reward into an offline RL algorithm which takes trajectories with total reward as input. More precisely, we split the outcome supervised data, use the first part to estimate the reward function via least squares, and then use this estimate to impute the missing rewards on the second part of the data. We summarize this basic transformation in the following algorithm.

---

**Algorithm 1** Offline Outcome-to-Process Transformation

---

1: **Input:** Offline dataset with total rewards $\mathcal{D}_O = \{(s_1, a_1, \cdots, s_H, a_H, R)\}$, Reward model class $\mathcal{R}$, Offline RL Algorithm $\mathfrak{A}$
2: Split $\mathcal{D}_O$ into two datasets $\mathcal{D}_O^1$ and $\mathcal{D}_O^2$ of equal size.
3: Compute $\widehat{r}$ by solving Eq. (3) with dataset $\mathcal{D}_O^1$ and the reward class $\mathcal{R}$.
4: Construct dataset $\mathcal{D}_P = \{(s_1, a_1, r_1, \cdots, s_H, a_H, r_H)$ from $\mathcal{D}_O^2$, where $\forall (s_1, a_1, \cdots, s_H, a_H, R) \in \mathcal{D}_O^2$,

$$r_h = \widehat{r}(s_h, a_h), \qquad \forall h \in [H].$$

5: Call algorithm $\mathfrak{A}$ with dataset $\mathcal{D}_P$ and output learned policy $\widehat{\pi}$.
6: **Output:** Learned policy $\widehat{\pi}$.

---

**Corollary 2.** *Fix a policy set $\Pi$ which contains the optimal policy $\pi^\star$. Suppose the offline RL algorithm $\mathfrak{A}$ always outputs a policy $\widehat{\pi} \in \Pi$ with error at most $\varepsilon_{\mathsf{alg}}$, i.e., $J(\pi^\star) - J(\widehat{\pi}) \leq \varepsilon_{\mathsf{alg}}$, with probability at least $1 - \delta$. Then the policy output by Algorithm 1 satisfies with probability at least $1 - 2\delta$,*

$$\max_\pi J(\pi) - J(\widehat{\pi})$$
$$\lesssim \varepsilon_{\mathsf{alg}} + H^{3/2} \cdot \sqrt{\frac{C_{\mathsf{sa}}(\Pi, \pi_{\mathsf{off}}) \cdot \log(|\mathcal{R}|/\delta)}{|\mathcal{D}_O|}},$$

*where $C_{\mathsf{sa}}(\Pi, \pi_{\mathsf{off}}) := \sup_{\pi \in \Pi} C_{\mathsf{sa}}(\pi, \pi_{\mathsf{off}})$.*

Notice that the bound in the above theorem suffers from all-policy concentrability, regardless of which algorithm $\mathfrak{A}$ is used with the transformed data. This occurs because the transformation fixes the learned reward model, requiring us to account for the distribution shift between $\pi_{\mathsf{off}}$ and the data-dependent policy $\widehat{\pi}$ which can be any policy in the policy class $\Pi$. This is a common issue in classical offline RL without specific methods like pessimism, particularly for the case of partial coverage. However, the concept of pessimism can also be applied to the outcome supervision setting in our paper, where we learn a specific reward model for each policy, as commonly done in the (process-supervision) offline RL literature (e.g., Xie et al., 2022a; Cheng et al., 2022; Uehara and Sun, 2021; Bhardwaj et al., 2023). Following this approach, we can transform model-based offline RL algorithms that use pessimism (Xie et al., 2022a; Bhardwaj et al., 2023) into algorithms that employ outcome supervision data. The sample complexity of these transformed algorithms scales with the single-policy concentrability $C_{\mathsf{sa}}(\pi^\star, \pi_{\mathsf{off}})$, which depends only on the optimal policy $\pi^\star$. We defer further details to Appendix C.

**Statistical Efficiency.** We now argue that there is no significant statistical edge for process supervision paradigm compared to the outcome supervision paradigm in the offline setting. The latter corresponds to standard offline RL problems (Levine et al., 2020; Jiang and Xie, 2024), for which a rich body of work exists analyzing sample complexity. Our "equivalence" argument primarily focus on coverage conditions, since different coverage notions (e.g., state-action-level vs. trajectory-level) can lead to exponential differences, as discussed in Section 2.3. While our results establish equivalence with respect to coverage conditions, we acknowledge they may still be subject to polynomial factors of $H$; removing such factors is an avenue for further research.

If we consider the worst-case scenario, it is easy to see that any algorithm which outputs an $\varepsilon$-optimal policy requires at least $\sup_\pi C_{\mathsf{sa}}(\pi, \pi_{\mathsf{off}})/\varepsilon^2$ number of samples. To see this, we may consider the two-armed bandit with action

$a_1$ and $a_2$, and $r(a_1) = \pm\varepsilon, r(a_2) = 0$, and policy $\pi = \mathrm{Unif}(\{a_1, a_2\})$. In the meantime, many classical offline RL algorithms, such as Fitted Q-Iteration (Antos et al., 2008; Munos and Szepesvári, 2008), the theoretical backbone of Deep Q-Network (DQN), require sample complexity that scales with $C_{\mathsf{sa}}(\Pi, \pi_{\mathsf{off}})/\varepsilon^2$ for obtaining an $\varepsilon$-optimal policy (Chen and Jiang, 2019; Xie and Jiang, 2020). Hence Corollary 2 provides a transformation with the same sample complexity as in these works (up to polynomial in horizon factors), when encountering outcome supervision reward data.

As for the instance-dependent case (corresponding to the single-policy coverage discussed in Section 2.3), the lower bound result in Xie et al. (2021b) shows that any algorithm requires at least $C_{\mathsf{sa}}(\pi^\star, \pi_{\mathsf{off}})/\varepsilon^2$ number of samples to output an $\varepsilon$-optimal policy, which only depends on the coverage of optimal policy $\pi^\star$. Recent offline RL algorithms (Xie et al., 2021a; Cheng et al., 2022; Uehara and Sun, 2021; Bhardwaj et al., 2023) indeed reach that sample complexity in terms of the single-policy concentrability $C_{\mathsf{sa}}(\pi^\star, \pi_{\mathsf{off}})$. Our results presented in Appendix C match the upper bound of these offline RL algorithms for the process supervision case and also enjoy the same sample complexity depending on the single-policy concentrability $C_{\mathsf{sa}}(\pi^\star, \pi_{\mathsf{off}})$.

### 3.2. Change of Trajectory Measure

The proof of Theorem 1 relies on the following key change of trajectory measure lemma, which states that changing the measure of trajectory returns can be done at the price of state-action concentrability, up to logarithmic and polynomial-in-horizon factors. This lemma will be used for bounding the error between the true reward model $r^\star$ and the learned reward model $\widehat{r}$. Thus, by setting $f = r^\star - \widehat{r}$, we only need to show that the expectation of the absolute value of $f(\tau) := \sum_{(s_h, a_h) \sim \tau} f(s_h, a_h)$ is small for $\tau \sim \pi$ when controlling under $\pi_{\mathsf{off}}$.

**Lemma 3.** *(Change of Trajectory Measure Lemma) For MDP $M = (\mathcal{S}, \mathcal{A}, T, f, H)$ with any function $f : \mathcal{S} \times \mathcal{A} \to [-1, 1]$, for any two policies $\pi$ and $\pi_{\mathsf{off}}$, we have*

$$\frac{\mathbb{E}_{\tau \sim \pi}[f(\tau)^2]}{\mathbb{E}_{\tau \sim \pi_{\mathsf{off}}}[f(\tau)^2]} \lesssim H^3 C_{\mathsf{sa}}(\pi, \pi_{\mathsf{off}}),$$

*where $C_{\mathsf{sa}}(\pi, \pi_{\mathsf{off}})$ is defined in Eq. (5). Additionally, the following holds without the extraneous log factors:*

$$\mathbb{E}_{\tau \sim \pi}[|f(\tau)|] \lesssim \sqrt{H^3 C_{\mathsf{sa}}(\pi, \pi_{\mathsf{off}}) \cdot \mathbb{E}_{\tau \sim \pi_{\mathsf{off}}}[f(\tau)^2]}.$$

This lemma reveals a perhaps surprising insight: when the squared sum of some state-action value functions $[\sum_{(s_h, a_h) \sim \tau} f(s_h, a_h)]^2$ is small under the off-policy trajectory distribution $\tau \sim \pi_{\mathsf{off}}$, we only need to account for state-action-level distribution shifts between $\pi_{\mathsf{off}}$ and $\pi$ to bound the same squared sum under $\tau \sim \pi$. This holds true

even though controlling such trajectory sums theoretically cannot prevent cases where individual terms have equal and large magnitude but opposite signs (i.e., where $|a| = |b| > 0$ but $a + b = 0$). We provide a proof sketch of Lemma 3 in the following. The detailed proof is deferred to Appendix B.1.

### 3.3. Proof Sketch of Lemma 3

In this section, we outline the key insights behind the proof of Lemma 3. The central observation is that controlling the trajectory-level variance of $f$ under a reference policy $\pi_{\text{off}}$ implies an automatic control of variance on prefixes and suffixes over the entire trajectory, with only a polynomial overhead in the horizon length $H$. This seemingly simple fact leads to perhaps surprisingly strong guarantees.

**Insight I: Trajectory-level bound controls the second moment on prefixes and suffixes.** At first glance, small value $|f(\tau)|$ over the entire trajectory $\tau$ does *not* obviously guarantee that the value of $f$ on either (i) every prefix $\tau_{1:h}$ or (ii) every suffix $\tau_{h+1:H}$ is small. In principle, large positive and large negative portions of a single trajectory could "cancel" each other out, resulting in a small overall sum $|f(\tau)| = |f(\tau_{1:h}) + f(\tau_{h+1:H})|$.

Crucially, however, thanks to the *Markov property*, we can argue that if $f$ has small second moment (under $\pi_{\text{off}}$) and if a state $s_h$ is visited sufficiently often by $\pi_{\text{off}}$, $f$ cannot have high variance on the prefix (leading up to $s_h$) and suffix (following $s_h$). Indeed, if the value of $f$ on the prefix (or suffix) has large variance, then conditioned on passing through $s_h$ that is visited sufficiently often by $\pi_{\text{off}}$, the value of $f$ on the entire trajectory also has large variance, which directly implies the large variance (hence, large second moment) of $f(\tau)$. Hence, even though the trajectory-level bound looks coarse, it forces each state $s_h$ to have relatively stable partial sums in both the prefix and suffix directions under $\pi_{\text{off}}$.

**Insight II: Layer-by-layer "locking" with only state-action coverage.** Next, we want to argue that if *all* states in a trajectory satisfy the above low-variance property (we call such states "good" states), then the reward of the entire trajectory cannot have large absolute value. We call this the "locking in" property here for brevity. In the following, we argue that "locking in" happens with high probability, even under policy $\pi$.

According to the earlier argument, "bad" states (opposite of "good" states) cannot have large visitation probability under $\pi_{\text{off}}$. Then, by the definition of $C_{\text{sa}}(\pi, \pi_{\text{off}})$, which upper bounds the probability ratio between $\pi$ and $\pi_{\text{off}}$ at any state, we conclude that such bad states also have low probability under $\pi$, up to a factor of $C_{\text{sa}}(\pi, \pi_{\text{off}})$. Thus, we avoid exponential blow-up over the horizon because we only "pay" for distribution shift at each individual $(s, a)$, rather

than for entire trajectories: $\mathbb{P}_{\tau \sim \pi}(\text{bad}) \leq C_{\text{sa}}(\pi, \pi_{\text{off}}) \cdot \mathbb{P}_{\tau \sim \pi_{\text{off}}}(\text{bad})$.

Hence, with high probability, $\pi$ visits only "good" states throughout its trajectory, ensuring that each layer $h$ "locks in" a small partial sum (as both its prefix and suffix have low variance).[1] When we stitch these layers from $h = 1$ to $h = H$, the entire sum $f(\tau)$ is guaranteed to have small absolute values.

### 3.4. Extension to Preference-Based Reinforcement Learning

In the previous section, we studied the statistical complexity of outcome supervision under the data format of Eq. (2), where the outcome reward is provided at the end of each trajectory. Preference-based reinforcement learning (e.g., Knox and Stone, 2008; Akrour et al., 2012; Wirth et al., 2017) represents another well-established paradigm that extends outcome supervision and is commonly employed for learning from human preferences (e.g., Griffith et al., 2013; Christiano et al., 2017; Stiennon et al., 2020; Ouyang et al., 2022).

Recent work on implicit reward modeling through single-step contrastive learning approaches (DPO; Rafailov et al., 2023) aims to eliminate the need for explicit reward modeling. While extensive prior work (e.g., Zhan et al., 2023; Liu et al., 2024; Song et al., 2024; Zhang et al., 2024) has focused on the sample complexity of these implicit reward modeling approaches, most existing bounds rely on trajectory-level change of measure. These results are also considered to depend on the trajectory concentrability under naive simplifications, which can grow exponentially with the horizon length (see detailed discussion in Section 2.3 and Section 3.2 of Xie et al. 2024).

In this section, we extend our main results to the preference-based reinforcement learning setting. As a direct application of our Change of Trajectory Measure Lemma (Lemma 3), we first provide a sample complexity bound of preference based RL which only scales with state-action concentrability instead of trajectory concentrability, a result applicable to standard explicit reward modeling approaches as well as implicit reward modeling approaches (i.e., DPO).

In preference-based RL, we suppose that for any trajectory $\tau$, the total reward along $\tau$ satisfies $r(\tau) \in [0, V_{\max}]$. To form the dataset $\mathcal{D}$ of preferences, the learner collects two reward-free trajectories $\tau = (s_1, a_1, s_2, a_2, \cdots, s_H, a_H)$ and $\tau' = (s'_1, a'_1, s'_2, a'_2, \cdots, s'_H, a'_H)$ according to policy $\pi_{\text{ref}}$, and receives the information about the order $(\tau_+, \tau_-)$ of $\tau, \tau'$, based on the preference $y \sim \mathbb{P}(\tau \succ \tau')$. We adopt

---

[1] Our formal proof also needs to consider the "good" state-action-state tuples, which are similar to "good" states but involve the $(s_h, a_h, s_{h+1})$ tuple. We omit the details here for brevity, and readers can refer to the full proof in Appendix B.1.

the Bradley-Terry model (Bradley and Terry, 1952):

$$\mathbb{P}(\tau \succ \tau') = \frac{\exp(r(\tau))}{\exp(r(\tau)) + \exp(r(\tau'))}. \quad (6)$$

The labeled preference dataset $\mathcal{D}$ consists of ordered samples $(\tau_+, \tau_-)$, where both trajectories are collected according to policy $\pi_{\mathsf{ref}}$ and labeled according to the Bradley-Terry model Eq. (6).

### 3.4.1. IMPROVED ANALYSIS OF PREFERENCE-BASED RL WITH EXPLICIT REWARD MODELING

We first provide the analysis for the case where an explicit reward modeling procedure is used for preference-based RL. Suppose the learner is given the preference-based dataset $\mathcal{D}_{\mathsf{pref}}$ and a reward class $\mathcal{R}$, where for every reward model $r \in \mathcal{R}$ and any trajectory $\tau$, $r(\tau) \in [0, V_{\max}]$. In the following result, an analogue of Theorem 1, we transform the preference-based dataset into the reward model via maximum likelihood rather than the method of least squares.

**Theorem 4.** *Suppose $\mathcal{D}_{\mathsf{pref}} = \{(\tau_+, \tau_-)\}$ contains i.i.d. pairs of sequences collected according to $\pi_{\mathsf{ref}}$ and ordered according to Eq. (6) with $r^\star \in \mathcal{R}$. Let*

$$\widehat{r} = \arg\min_{r \in \mathcal{R}} \sum_{(\tau_+, \tau_-) \in \mathcal{D}_{\mathsf{pref}}} \log \sigma(r(\tau_+) - r(\tau_-))$$

*be the maximum likelihood estimate. With probability at least $1 - \delta$, for any two policies $\pi, \pi' \in \Pi$, it holds that*

$$\mathbb{E}_{\tau \sim \pi, \tau' \sim \pi'}[|r^\star(\tau) - r^\star(\tau') - \widehat{r}(\tau) + \widehat{r}(\tau')|]$$
$$\lesssim H^{3/2} V_{\max} e^{2V_{\max}}$$
$$\cdot \sqrt{\frac{(C_{\mathsf{sa}}(\pi, \pi_{\mathsf{ref}}) \vee C_{\mathsf{sa}}(\pi', \pi_{\mathsf{ref}})) \log(|\Pi|/\delta)}{|\mathcal{D}|}}. \quad (7)$$

The proof of Theorem 4 is deferred to Appendix B.3. Notice that with Eq. (7), if we could obtain $\widehat{\pi}$ as the optimal policy under reward model $\widehat{r}$ (otherwise we can just pay for the sub-optimality as in Corollary 2), then with high probability $\widehat{\pi}$ is a near-optimal policy. To see this, we choose $\pi = \widehat{\pi} = \arg\max_\pi J_{\widehat{r}}(\widehat{\pi})$ and $\pi' = \pi^\star = \arg\max_\pi J(\widehat{\pi})$, then with probability at least $1 - \delta$,

$$J(\pi^\star) - J(\widehat{\pi})$$
$$\leq J(\pi^\star) - J(\widehat{\pi}) - J_{\widehat{r}}(\pi^\star) + J_{\widehat{r}}(\widehat{\pi})$$
$$= \mathbb{E}_{\tau \sim \widehat{\pi}, \tau' \sim \pi^\star}[r^\star(\tau') - r^\star(\tau) - \widehat{r}(\tau') + \widehat{r}(\tau)]$$
$$\leq \mathbb{E}_{\tau \sim \widehat{\pi}, \tau' \sim \pi^\star}[|r^\star(\tau') - r^\star(\tau) - \widehat{r}(\tau') + \widehat{r}(\tau)|]$$
$$\lesssim H^{3/2} V_{\max} e^{2V_{\max}}$$
$$\cdot \sqrt{\frac{(C_{\mathsf{sa}}(\widehat{\pi}, \pi_{\mathsf{ref}}) \vee C_{\mathsf{sa}}(\pi^\star, \pi_{\mathsf{ref}})) \log(|\Pi|/\delta)}{|\mathcal{D}|}}$$
$$\leq H^{3/2} V_{\max} e^{2V_{\max}} \cdot \sqrt{\frac{C_{\mathsf{sa}}(\Pi, \pi_{\mathsf{off}}) \log(|\Pi|/\delta)}{|\mathcal{D}|}},$$

Therefore, as we acquire enough samples, $J(\pi^\star) - J(\widehat{\pi})$ converges to zero with high probability, implying that $\widehat{\pi}$ is a near-optimal policy. This convergence requires the same condition of bounded state-action concentrability as in Corollary 2.

### 3.4.2. IMPROVED ANALYSIS OF DPO ALGORITHM

We now extend our main results to the implicit reward modeling setting and analyze the sample complexity of the DPO algorithm (Rafailov et al., 2023). DPO is a popular implicit reward algorithm that converts the two-step process of reward modeling and policy optimization into a single-step contrastive learning problem. DPO is commonly used in the token-level setup of LLMs, where actions (tokens) are directly appended to states (contexts) (Rafailov et al., 2023; 2024). In this case, the state-action concentrability coefficient essentially reduces to trajectory-level concentrability, as the last state is contains the trajectory. However, recent work indicates that DPO-style algorithms are applicable beyond the token-level setup, e.g., in environments with deterministic transition dynamics but still Markovian states, e.g., in robotics (Hejna et al., 2023; Xie et al., 2024). In these settings, our bounds with only state-action concentrability can be substantially tighter than existing trajectory-level ones.

Following Xie et al. (2024), we assume deterministic ground-truth transition dynamics and consider the following KL-regularized objective (Xiong et al., 2023; Ye et al., 2024; Xie et al., 2024): for some positive number $\beta$,

$$\mathcal{J}_\beta(\pi) := J_r(\pi) - \beta D_{\mathsf{KL}}(\pi(\tau) \| \pi_{\mathsf{ref}}(\tau))$$
$$= \mathbb{E}_{\tau \sim \pi}\left[r(\tau) - \beta \log \frac{\pi(\tau)}{\pi_{\mathsf{ref}}(\tau)}\right]. \quad (8)$$

The policy $\pi_\beta^\star$ which maximizes $\mathcal{J}_\beta(\pi)$ in Eq. (8) satisfies $\pi_\beta^\star(\tau) \propto \pi_{\mathsf{ref}}(\tau) \exp(r^\star(\tau)/\beta)$ for any trajectory $\tau$. It is easy to verify that $\pi_\beta^\star$ is a Markov policy. We assume the learner has access to a Markov policy class $\Pi \subset \mathcal{S}^\mathcal{A}$, and aims to find a policy $\widehat{\pi}$ that is nearly optimal with respect to the policy class $\Pi$, i.e.

$$\max_{\pi \in \Pi} \mathcal{J}_\beta(\pi) - \mathcal{J}_\beta(\widehat{\pi}) \leq \varepsilon.$$

The DPO algorithm (Rafailov et al., 2023) takes the dataset $\mathcal{D} = \{(\tau_+, \tau_-)\}$ as input, and outputs the policy $\widehat{\pi} \in \Pi$ which maximizes the log likelihood, i.e., $\widehat{\pi}$ is obtained by solving

$$\arg\min_{\pi \in \Pi}\left\{\sum_{(\tau_+, \tau_-) \in \mathcal{D}} \log\left[\sigma\left(\beta \log \frac{\pi(\tau_+)}{\pi_{\mathsf{ref}}(\tau_+)} - \beta \log \frac{\pi(\tau_-)}{\pi_{\mathsf{ref}}(\tau_-)}\right)\right]\right\}$$
$$(9)$$

In the following, we provide a refined analysis of the sample complexity of this algorithm. We first make the following assumptions.

**Assumption 1** (Policy Realizability). *The optimal policy is in the policy class $\Pi$, i.e., $\pi_\beta^\star \in \Pi$.*

Policy realizability is a minimal assumption for sample-efficient reinforcement learning and is necessary for establishing many standard results (Agarwal et al., 2019; Lattimore and Szepesvári, 2020; Foster and Rakhlin, 2023).

In addition, we make the following assumptions of bounded trajectory concentrability.

**Assumption 2.** *For any policy $\pi \in \Pi$ and trajectory $\tau$, $\left| \log \frac{\pi(\tau)}{\pi_{\mathrm{ref}}(\tau)} \right| \leq \frac{V_{\max}}{\beta}$.*

This assumption is commonly used in the literature on implicit reward modeling approaches for preference-based reinforcement learning (e.g., Rosset et al., 2024; Xie et al., 2024). The intuition behind this assumption is that we treat $\log \frac{\pi(\tau)}{\pi_{\mathrm{ref}}(\tau)}$ as an implicit value function,[2] so that this assumption essentially plays the same role as bounded value functions in the analysis.

If the above assumptions hold, we have the following upper bound on the sample complexity of the DPO algorithm in terms of the state-action concentrability.

**Theorem 5.** *Suppose Assumptions 1 and 2 hold, with probability at least $1 - \delta$, the output $\widehat{\pi}$ of the DPO algorithm in Eq. (9) satisfies*

$$\mathcal{J}_\beta(\pi_\beta^\star) - \mathcal{J}_\beta(\widehat{\pi})$$

$$\lesssim H^{3/2} V_{\max} e^{2V_{\max}} \cdot \sqrt{\frac{\sup_{\pi \in \Pi} C_{\mathsf{sa}}(\pi, \pi_{\mathsf{off}}) \log(|\Pi|/\delta)}{|\mathcal{D}|}}.$$

The proof of Theorem 5 is deferred to Appendix B.3. Note that, the exponential dependence on the reward range $V_{\max}$ is a fundamental characteristic of the Bradley-Terry model, not an artifact of our analysis. Indeed, this exponential dependence appears consistently across the theoretical literature on RLHF (e.g., Zhu et al., 2023; Zhan et al., 2023; Rosset et al., 2024; Xie et al., 2024; Das et al., 2024).

## 4. Advantage Function Learning with Rollouts

In this section, we analyze a common empirical strategy for converting outcome-supervised data into process supervision by leveraging online rollouts. The central observation is that, given access to an environment that returns final outcomes, one can initiate rollouts from individual state-action pairs and use the resulting outcomes to approximate their "quality." Multiple works have adopted variations of this idea, relying on Q-functions (e.g., Wang et al., 2024), advantage functions (e.g., Setlur et al., 2024), or other specialized value estimators (e.g., Luo et al., 2024).

---

[2] In fact, $\log \pi(\tau)/\pi_{\mathrm{ref}}(\tau)$ corresponds to a more complex combination of value functions and rewards. We refer readers to Watson et al. (2023); Rafailov et al. (2024); Xie et al. (2024) for further details.

Although these methods have demonstrated empirical promise, their theoretical properties remain relatively unexplored. Establishing a theoretical foundation could reveal the assumptions and conditions under which these methods are effective and enable principled comparisons to alternative reward modeling approaches. In what follows, we present (to our knowledge) the first theoretical study of advantage-based reward learning with online rollouts. We show that the advantage function of *any* policy can serve as a valid process-based reward model, recovering the same optimal policy as the original environment. By contrast, we also prove a lower bound indicating that simply using the Q-function can fail: in certain cases, the Q-function-based reward model produces suboptimal or undesired policies.

### 4.1. Algorithm and Upper Bounds

For MDP is $M = (\mathcal{S}, \mathcal{A}, P, r, H)$, suppose the transition model $P$ is known to the learner, but the reward model $r$ is unknown. For any given policy $\mu$, we define the advantage function $A^\mu : \mathcal{S} \times \mathcal{A} \to \mathbb{R}$ as

$$A^\mu(s, a) = Q_h^\mu(s, a) - V_h^\mu(s), \ \forall h \in [H], s \in \mathcal{S}_h, a \in \mathcal{A}, \quad (10)$$

where $Q_h^\mu$ and $V_h^\mu$ denote the $Q$-function and value function of policy $\mu$.

Setlur et al. (2024) find an approximation $\widehat{r}$ to $A^\mu$ and then invoke a policy gradient algorithm for the MDP with transition model $P$ and reward function $\widehat{r}$, yielding a policy $\widehat{\pi}$. We provide a theoretical guarantee for this approach by showing that the performance gap of $\widehat{\pi}$ to the optimal policy can be upper bounded in terms of the error $\varepsilon_{\mathsf{stat}}$ of approximating the advantage function and the error $\varepsilon_{\mathsf{alg}}$ of optimizing the policy. Before stating the result, we define the concentrability coefficient with respect to distribution $\nu$,

$$C_{\mathsf{sa}}(\nu) := \sup_\pi \sup_{s \in \mathcal{S}, a \in \mathcal{A}} \frac{d^\pi(s, a)}{\nu(s, a)}, \quad (11)$$

where the outer supremum is over all possible policies.

**Theorem 6.** *Suppose there exists some policy $\mu$ and distribution $\nu \in \Delta(\mathcal{S} \times \mathcal{A})$ such that*

$$\mathbb{E}_{(s,a)\sim\nu} \left[ |\widehat{r}(s, a) - A^\mu(s, a)|^2 \right] \leq \varepsilon_{\mathsf{stat}}. \quad (12)$$

*If policy $\widehat{\pi}$ satisfies*

$$\max_\pi J_{\widehat{r}}(\pi) - J_{\widehat{r}}(\widehat{\pi}) \leq \varepsilon_{\mathsf{alg}}, \quad (13)$$

*then it also satisfies*

$$\max_\pi J(\pi) - J(\widehat{\pi}) \leq 2H \sqrt{C_{\mathsf{sa}}(\nu)} \cdot \varepsilon_{\mathsf{stat}} + \varepsilon_{\mathsf{alg}}.$$

The proof of Theorem 6 is deferred to Appendix D. Intuitively, the proof follows from the performance difference lemma (Kakade and Langford, 2002), which states

that for any policies $\pi$ and $\mu$: $J(\pi) - J(\mu) = H \cdot \mathbb{E}_{(s_h, a_h) \sim d^\pi}[A^\mu(s_h, a_h)]$. This implies that maximizing $J_{A^\mu}$ (treating $A^\mu$ as the reward function) is equivalent to maximizing $J$, since they differ only by the constant term $J(\mu)$. Therefore, both optimization problems yield the same optimal policy.

There are several ways of obtaining an estimate $\widehat{r}$ of the advantage function $A^\mu$ that satisfies Eq. (12). One commonly used approach is Monte-Carlo sampling, for instance as in Setlur et al. (2024). In detail, this approach first collects a dataset $\mathcal{D}$ of data $(s, a, \widehat{A})$, where $(s, a) \in \mathcal{S} \times \mathcal{A}$ and $\widehat{A}$ is calculated via rollout from $(s, a)$ under policy $\mu$, serving as an approximation to the advantage function of policy $\mu$ at $(s, a)$. Next, we fit a reward function $\widehat{r}$ in a reward class to the dataset $\mathcal{D}$. Then, as long as the reward class realizes the ground truth advantage function $A^\mu$, the reward function $\widehat{r}$ satisfies Eq. (12) with high probability.

### 4.2. Lower Bound on Failure of Using Q-Functions

Theorem 6 indicates that the MDP with reward function set to be the advantage function of any policy $\mu$ has the same best policy as the original MDPs. One may wonder whether the same holds for the $Q$-function as well. In this section, we disprove this by providing a hard MDP with best policy $\pi^\star$, and a policy $\mu$, so that the best policy of the MDP with reward function $Q^\mu$ is not $\pi^\star$.

**Theorem 7.** *There exists an MDP $M = (\mathcal{S}, \mathcal{A}, P, r, H)$, and a policy $\mu \in \mathcal{A}^\mathcal{S}$, such that*

$$\max_\pi J_r(\pi) - J_r(\widehat{\pi}) \geq \frac{1}{3},$$

*for $\widehat{\pi} = \arg\max_\pi J_{Q^\mu}(\pi)$. Here $Q^\mu : \mathcal{S} \times \mathcal{A} \to \mathbb{R}$ is the $Q$-function of MDP $M$.*

The proof of Theorem 7 is deferred to Appendix D.

Theorem 7 indicates that the widely used approach in large language model training, whereby an approximate $Q$-function is learned in place of the reward model, is theoretically incorrect and possibly outputs undesired policies. In contrast, using the advantage function as the reward model is theoretically justified.

## 5. Conclusion

In this paper, we present a way of transforming data in the setting of outcome supervision into the data in the setting of process supervision. This transformation enables us to design offline algorithms in the outcome supervision model from the large pool of algorithms that use process supervision. This transformation extends to preference-based algorithms such as DPO. Beyond this transformation, we also provide theoretical guarantees for algorithms using an approximate advantage function as the reward function.

While our transformation scheme works for most of the offline algorithms, the theoretical guarantees require that the outcome supervision data are collected offline. How to construct similar transformation for online data or online algorithms is left for future work.

## Acknowledgements

ZJ and AR acknowledge support of the Simons Foundation and the NSF through awards DMS-2031883 and PHY-2019786, as well as from the ARO through award W911NF-21-1-0328.

## Impact Statement

This work contributes to advancing the field of Machine Learning through theoretical analysis and insights. While our research focuses on foundational understanding rather than direct applications, we acknowledge that theoretical advancements in machine learning can have broad societal implications. However, given the theoretical nature of this work, we believe these potential impacts do not warrant specific discussion in this context.

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

# Appendix

## A. Related Work

**Process vs. Outcome Supervision.** Our work is motivated by the empirical effectiveness of process supervision over outcome supervision, particularly in language model reasoning tasks (Cobbe et al., 2021; Uesato et al., 2022; Lightman et al., 2023). To address the challenges of cost and scalability in obtaining human-annotated process labels, recent approaches (Wang et al., 2024; Luo et al., 2024; Setlur et al., 2024) have developed automated methods to generate process supervision from outcome-based signals, leveraging Q-functions and advantage functions under specific policies. When data is provided in the form of preferences, outcome supervision is sometimes conducted with implicit rewards, as seen in works such as Direct Preference Optimization (Rafailov et al., 2023; Lambert et al., 2024; Zhong et al., 2024; Yuan et al., 2024).

**RL with Trajectory Feedback.** A closely related line of theoretical work is reinforcement learning with trajectory feedback or aggregate bandit feedback (or, bandit and semi-bandit feedback) (Neu and Bartók, 2013; Efroni et al., 2021; Chatterji et al., 2021; Chen et al., 2022; Cassel et al., 2024; Lancewicki and Mansour, 2025), where the learner only receives trajectory-level feedback at the end of each episode. This line of work also includes preference-based RL (Pacchiano et al., 2021; Zhu et al., 2023; Wu and Sun, 2023; Zhan et al., 2023), which operates on trajectory-level pair preferences. While most existing works in this area focus on online exploration settings, our paper investigates offline learning and analyzes the statistical relationship between process (step-level) and outcome (trajectory-level) feedback.

**Offline Reinforcement Learning.** Our work is most closely related to offline (batch) reinforcement learning in the classical reinforcement learning literature. The paradigm of reinforcement learning with process-supervised data is essentially an offline RL problem, where a rich body of existing theoretical results (e.g., Munos, 2003; Antos et al., 2008; Farahmand et al., 2010; Chen and Jiang, 2019; Xie and Jiang, 2020; Jin et al., 2021; Xie and Jiang, 2021; Xie et al., 2021a; Uehara and Sun, 2021; Cheng et al., 2022; Xie et al., 2022a; Bhardwaj et al., 2023) can be applied to our paper, either directly for the process supervision case, or serving as subroutines in our Algorithm 1 for the outcome supervision case. Within these results, Chen and Jiang (2019); Xie and Jiang (2020) develop model-free algorithms under all-policy coverage conditions, while Xie and Jiang (2021) proposes a model-free approach requiring only realizability assumptions but with stronger coverage requirements. Xie et al. (2021a); Cheng et al. (2022) investigate model-free offline RL under partial coverage settings, and Uehara and Sun (2021); Xie et al. (2022a); Bhardwaj et al. (2023) address model-based offline RL with partial coverage.

**Off-Policy Evaluation.** Our work also connects to the rich literature on off-policy evaluation (OPE) in reinforcement learning. A central challenge in OPE is the change of measure problem, where extensive research (Liu et al., 2018; Xie et al., 2019; Nachum et al., 2019; Uehara et al., 2020) has investigated the significant distinction between state-action coverage and trajectory coverage conditions. These findings highlight the significance of our main results, particularly our change of trajectory measure lemma.

**Reward Shaping and Internal Rewards.** A related but distinct line of research focuses on augmenting sparse reward functions to improve learning efficiency. Reward shaping techniques have been extensively employed as a method for providing denser learning signals while preserving optimal policies (e.g., Ng et al., 1999; Trott et al., 2019; Gupta et al., 2022). Similarly, intrinsic rewards based on prediction errors of environment dynamics (Pathak et al., 2017) or random networks (Burda et al., 2018) have been proposed to tackle sparse reward settings. However, these approaches differ fundamentally from our work in their objectives – while reward shaping and intrinsic rewards aim to improve exploration in online RL by modifying the reward landscape, our analysis focuses on the statistical properties of learning in the offline setting where the data distribution is fixed.

## B. Missing Proofs in Section 3

### B.1. Proof of Change of Trajectory Measure Lemma

***Proof of Lemma 3.*** In the following proof, for any trajectory $\tau = (s_1, a_1, s_2, a_2, \cdots, s_H, a_H)$ and $1 \le u \le v \le H$, we use $\tau_{u:v}$ to denote the partial trajectory $(s_u, a_u, s_{u+1}, a_{u+1}, \cdots, s_v, a_v)$. We use $f(\tau_{u:v}) := \sum_{h=u}^{v} f(s_h, a_h)$ to denote the cumulative reward in this segment. Without loss of generality we assume for any trajectory $\tau$, $f(\tau) \in [-1, 1]$. Since the state space of the MDP is layered, we write $\{s_h \in \tau\}$ for $s_h \in \mathcal{S}_h$ to denote the event that $s_h$ is the time-$h$ element of the trajectory $\tau$. We let

$$L(\eta) = \mathbf{P}_{\tau \sim \pi_{\text{off}}} \left[ |f(\tau)| \ge \eta \right]$$

for every $\eta > 0$. For every real number $r \in \mathbb{R}$ and $\eta, p > 0$, we define the following sets:

$$\mathcal{S}_h^{\uparrow}(r, \eta, p) := \left\{ s_h \in \mathcal{S}_h : \mathbf{P}_{\tau \sim \pi_{\text{off}}} \left( |r - f(\tau_{h:H})| \leq \eta \,\Big|\, s_h \in \tau \right) \geq 1 - p \right\},$$

$$\text{and} \quad \mathcal{S}_h^{\downarrow}(r, \eta, p) := \left\{ s_h \in \mathcal{S}_h : \mathbf{P}_{\tau \sim \pi_{\text{off}}} \left( |r - f(\tau_{1:h-1})| \leq \eta \,\Big|\, s_h \in \tau \right) \geq 1 - p \right\}.$$

Intuitively, $\mathcal{S}_h^{\downarrow}(r, \eta, p)$ denotes subset of states in $\mathcal{S}_h$ where, conditionally on arriving at that state under policy $\pi_{\text{off}}$, with probability at least $1 - p$, the total reward collected in the first $(h - 1)$ steps is $\eta$-close to $r$. Similarly, $\mathcal{S}_h^{\uparrow}(r, \eta, p)$ denotes subset of states in $\mathcal{S}_h$ where, conditionally on arriving at that state under policy $\pi_{\text{off}}$, the probability that the total reward collected in the last $(H - h + 1)$ steps is $\eta$-close to $r$ is at least $1 - p$.

We further define sets

$$\mathcal{S}_h^{\uparrow}(\eta, p) = \cup_{r \in \mathbb{R}} \mathcal{S}_h^{\uparrow}(r, \eta, p), \quad \mathcal{S}_h^{\downarrow}(\eta, p) = \cup_{r \in \mathbb{R}} \mathcal{S}_h^{\downarrow}(r, \eta, p).$$

We can now upper bound the occupancy measure of those states outside the set $\mathcal{S}_h^{\uparrow}(\eta, p)$ or $\mathcal{S}_h^{\downarrow}(\eta, p)$. This property is summarized in the following claim:

**Claim 1.** We have the following upper bounds on the occupancy measure outside $\mathcal{S}_h^{\uparrow}(\eta, p)$ and $\mathcal{S}_h^{\downarrow}(\eta, p)$:

$$\mathbf{P}_{\tau \sim \pi_{\text{off}}}(\tau \cap \mathcal{S}_h \not\subset \mathcal{S}_h^{\uparrow}(\eta, p)) \leq \frac{L(\eta)}{p} \quad \text{and} \quad \mathbf{P}_{\tau \sim \pi_{\text{off}}}(\tau \cap \mathcal{S}_h \not\subset \mathcal{S}_h^{\downarrow}(\eta, p)) \leq \frac{L(\eta)}{p}. \tag{14}$$

**Proof.** We only prove the first inequality, as the proof of the second inequality is similar. Notice that for any state $s_h \in \mathcal{S}_h \backslash \mathcal{S}_h^{\uparrow}(\eta, p)$, according to the definition of $\mathcal{S}_h^{\uparrow}(\eta, p)$ we have for any $r \in \mathbb{R}$,

$$\mathbf{P}_{\tau \sim \pi_{\text{off}}} \left( |r - f(\tau_{h:H})| \leq \eta \,\Big|\, s_h \in \tau \right) < 1 - p.$$

According to the Markov property, when sampling $\tau \sim \pi_{\text{off}}$, $\tau_{1:h-1} \perp\!\!\!\perp \tau_{h:H}$ conditioned on $s_h \in \tau$, which implies

$$\mathbf{P}_{\tau \sim \pi_{\text{off}}} \left( |f(\tau)| \leq \eta \,\Big|\, s_h \in \tau \right)$$
$$= \mathbb{E}_{\tau \sim \pi_{\text{off}}} \left[ \mathbf{P}_{\tau \sim \pi_{\text{off}}} \left[ |f(\tau_{h:H}) - (-f(\tau_{1:h-1}))| \leq \eta \,\Big|\, \tau_{1:h-1}, s_h \in \tau \right] \,\Big|\, s_h \in \tau \right] \leq 1 - p.$$

Hence we have

$$L(\eta) = \mathbf{P}_{\tau \sim \pi_{\text{off}}} \left[ |f(\tau)| \geq \eta \right] \geq \mathbf{P}_{\tau \sim \pi_{\text{off}}}(\tau \cap \mathcal{S}_h \not\subset \mathcal{S}_h^{\uparrow}(\eta, p)) \cdot p,$$

which implies the first inequality of Eq. (14). The second inequality of Eq. (14) follows similarly. $\square$

Next, for every real number $r \in \mathbb{R}$ and $\eta, p > 0$, we define the set

$$\mathcal{S}_h(r, \eta, p) := \left\{ s_h \in \mathcal{S}_h : \mathbf{P}_{\tau \sim \pi_{\text{off}}} \left( |r - f(\tau_{1:h-1})| \leq \eta \text{ and } |r + f(\tau_{h:H})| \leq \eta \,\Big|\, s_h \in \tau \right) \geq 1 - p \right\}.$$

This set denotes subset of states in $\mathcal{S}_h$ where conditioned on arriving at that state under policy $\pi_{\text{off}}$, the probability that the first $(h - 1)$ steps' total reward is $\eta$-close to $r$ and also the last $(H - h + 1)$ steps' total reward is $\eta$-close to $-r$ is less than $p$. We further define the set

$$\mathcal{S}_h(\eta, p) = \cup_{r \in \mathbb{R}} \mathcal{S}_h(r, \eta, p),$$

Then we have the following claim, which shows that the occupancy measure of states outside $\mathcal{S}_h(\eta, p)$ is also upper bounded:

**Claim 2.** We have the following upper bounds on the occupancy measure outside $\mathcal{S}_h(\eta, p)$:

$$\mathbf{P}_{\tau \sim \pi_{\text{off}}}(\tau \cap \mathcal{S}_h \notin \mathcal{S}_h(\eta, p)) \leq \frac{L(2\eta/3)}{1 - p} + \frac{4L(\eta/3)}{p}. \tag{15}$$

**Proof.** For state $s_h \in \mathcal{S}_h \backslash \mathcal{S}_h(\eta, p)$ but $s_h \in \mathcal{S}_h^{\uparrow}(\eta/3, p/2) \cap \mathcal{S}_h^{\downarrow}(\eta/3, p/2)$, there exists some $r^{\uparrow}(s_h) \in \mathbb{R}$ and $r^{\downarrow}(s_h) \in \mathbb{R}$ such that

$$\mathbf{P}_{\tau \sim \pi_{\text{off}}} \left( \left| f(s_h)^{\downarrow} - f(\tau_{1:h-1}) \right| \leq \frac{\eta}{3} \,\Big|\, s_h \in \tau \right) \geq 1 - \frac{p}{2} \text{ and } \mathbf{P}_{\tau \sim \pi_{\text{off}}} \left( \left| f(s_h)^{\uparrow} - f(\tau_{h:H}) \right| \leq \frac{\eta}{3} \,\Big|\, s_h \in \tau \right) \geq 1 - \frac{p}{2}.$$

By union bound we have that for any such $s_h$,

$$\mathbf{P}_{\tau \sim \pi_{\text{off}}} \left( \left| r^{\downarrow}(s_h) - f(\tau_{1:h-1}) \right| \leq \frac{\eta}{3} \quad \text{and} \quad \left| r^{\uparrow}(s_h) - f(\tau_{h:H}) \right| \leq \frac{\eta}{3} \,\Big|\, s_h \in \tau \right) \geq 1 - p.$$

If for some $s_h \in \mathcal{S}_h \backslash \mathcal{S}_h(\eta, p)$, we have $|r^{\downarrow}(s_h) + r^{\uparrow}(s_h)| \leq \frac{4\eta}{3}$, then by letting $r = \frac{r^{\downarrow}(s_h) - r^{\uparrow}(s_h)}{2}$, we obtain

$$\mathbf{P}_{\tau \sim \pi_{\text{off}}} \left( |r - f(\tau_{1:h-1})| \leq \eta \text{ and } |r + f(\tau_{h:H})| \leq \eta \,\Big|\, s_h \in \tau \right)$$
$$\geq \mathbf{P}_{\tau \sim \pi_{\text{off}}} \left( \left| r^{\downarrow}(s_h) - f(\tau_{1:h-1}) \right| \leq \frac{\eta}{3} \text{ and } \left| r^{\uparrow}(s_h) - f(\tau_{h:H}) \right| \leq \frac{\eta}{3} \,\Big|\, s_h \in \tau \right)$$
$$\geq 1 - p.$$

This contradicts the definition of $\mathcal{S}_h \backslash \mathcal{S}_h(\eta, p)$. Hence for any $s_h \in (\mathcal{S}_h \backslash \mathcal{S}_h(\eta, p)) \cap (\mathcal{S}_h^{\uparrow}(\eta/3, p/2) \cap \mathcal{S}_h^{\downarrow}(\eta/3, p/2))$, we always have $|r^{\downarrow}(s_h) + r^{\uparrow}(s_h)| > \frac{4\eta}{3}$, which implies that

$$\mathbf{P}_{\tau \sim \pi_{\text{off}}} \left( |f(\tau)| \geq \frac{2\eta}{3} \,\Big|\, s_h \in \tau \right) \geq \mathbf{P}_{\tau \sim \pi_{\text{off}}} \left( \left| r^{\downarrow}(s_h) - f(\tau_{1:h-1}) \right| \leq \frac{\eta}{3} \text{ and } \left| r^{\uparrow}(s_h) - f(\tau_{h:H}) \right| \leq \frac{\eta}{3} \,\Big|\, s_h \in \tau \right)$$
$$\geq 1 - p.$$

Further notice that

$$L(2\eta/3) = \mathbf{P}_{\tau \sim \pi_{\text{off}}} \left( |f(\tau)| \geq 2\eta/3 \right)$$
$$\geq \mathbf{P}_{\tau \sim \pi_{\text{off}}} \left( \tau \cap \mathcal{S}_h \subset (\mathcal{S}_h \backslash \mathcal{S}_h(\eta, p)) \cap (\mathcal{S}_h^{\uparrow}(\eta/3, p/2) \cap \mathcal{S}_h^{\downarrow}(\eta/3, p/2)) \right) \cdot (1 - p).$$

Hence we obtain

$$\mathbf{P}_{\tau \sim \pi_{\text{off}}} \left( \tau \cap \mathcal{S}_h \subset (\mathcal{S}_h \backslash \mathcal{S}_h(\eta, p)) \cap (\mathcal{S}_h^{\uparrow}(\eta/3, p/2) \cap \mathcal{S}_h^{\downarrow}(\eta/3, p/2)) \right) \leq \frac{L(2\eta/3)}{1 - p}.$$

Additionally, according to our previous claim, we have

$$\mathbf{P}_{\tau \sim \pi_{\text{off}}} \left( \tau \cap \mathcal{S}_h \subset \mathcal{S}_h^{\uparrow}(\eta/3, p/2) \right) \leq \frac{2L(\eta/3)}{p} \quad \text{and} \quad \mathbf{P}_{\tau \sim \pi_{\text{off}}} \left( \tau \cap \mathcal{S}_h \subset \mathcal{S}_h^{\downarrow}(\eta/3, p/2) \right) \leq \frac{2L(\eta/3)}{p}.$$

Combining the above three inequalities, we obtain that

$$\mathbf{P}_{\tau \sim \pi_{\text{off}}} \left( \tau \cap \mathcal{S}_h \not\subset \mathcal{S}_h(\eta, p) \right) \leq \frac{L(2\eta/3)}{1 - p} + \frac{4L(\eta/3)}{p},$$

and Eq. (15) is verified. $\qquad \square$

We next define the following sets of "good" state-action-state tuples:

$$\mathcal{U}_h := \{(s_h, a_h, s_{h+1}) : s_h \in \mathcal{S}_h, a_h \in \mathcal{A}_h, s_{h+1} \in \mathcal{S}_{h+1},$$
$$\exists r, r' \in \mathbb{R} \text{ such that } s_h \in \mathcal{S}_h(r, \eta, p), s_{h+1} \in \mathcal{S}_{h+1}(r', \eta, p) \text{ and } |f(s_h, a_h) + r - r'| \leq 3\eta\}$$

and also "bad" state-action-state tuples:

$$\mathcal{U}_h' := \{(s_h, a_h, s_{h+1}) : s_h \in \mathcal{S}_h, a_h \in \mathcal{A}_h, s_{h+1} \in \mathcal{S}_{h+1},$$
$$\forall r, r' \in \mathbb{R} \text{ such that } s_h \in \mathcal{S}_h(r, \eta, p), s_{h+1} \in \mathcal{S}_{h+1}(r', \eta, p) \text{ and } |f(s_h, a_h) + r - r'| \geq 3\eta\}.$$

We will show that under policy $\pi_{\text{off}}$, the encountered state-action-state pairs are 'good', i.e., belong to $\mathcal{U}_h$, with high probability. Notice that the complement of set $\mathcal{U}_h$ satisfies

$$(\mathcal{U}_h)^c \subset \mathcal{U}'_h \cup \{(s_h, a_h, s_{h+1}) : s_h \in \mathcal{S}_h \backslash \mathcal{S}_h(\eta, p)\} \cup \{(s_h, a_h, s_{h+1}) : s_{h+1} \in \mathcal{S}_{h+1} \backslash \mathcal{S}_{h+1}(\eta, p)\}, \tag{16}$$

and according to the last claim we have

$$\mathbf{P}_{\tau \sim \pi_{\text{off}}}(\tau \cap \mathcal{S}_h \not\subset \mathcal{S}_h(\eta, p)) \le \frac{L(2\eta/3)}{1-p} + \frac{4L(\eta/3)}{p}$$

$$\text{and} \quad \mathbf{P}_{\tau \sim \pi_{\text{off}}}(\tau \cap \mathcal{S}_{h+1} \not\subset \mathcal{S}_{h+1}(\eta, p)) \le \frac{L(2\eta/3)}{1-p} + \frac{4L(\eta/3)}{p}. \tag{17}$$

We next upper bound $\mathbf{P}_{\tau \sim \pi_{\text{off}}}(\tau \cap \mathcal{U}'_h \ne \emptyset)$, which is summarized into the following claim.

**Claim 3.** We have the following upper bounds on the occupancy measure of $\mathcal{U}'_h$:

$$\mathbf{P}_{\tau \sim \pi_{\text{off}}}(\tau \cap \mathcal{U}'_h \ne \emptyset) \le \frac{L(\eta)}{1-2p}. \tag{18}$$

**Proof.** According to the Markov property, we have

$$(s_1, a_1, \cdots, s_{h-1}, a_{h-1}) \perp\!\!\!\perp (a_h, s_{h+1}) \quad \text{conditioned on } s_h$$
$$\text{and} \quad (s_{h+1}, a_{h+1}, \cdots, s_H, a_H) \perp\!\!\!\perp (s_h, a_h) \quad \text{conditioned on } s_{h+1}.$$

Hence for any $(s_h, a_h, s_{h+1}) \in \mathcal{U}'_h$, there exists $r$ and $r'$ such that $s_h \in \mathcal{S}_h(r, \eta, p)$ and $s_{h+1} \in \mathcal{S}_{h+1}(r', \eta, p)$, which implies

$$\mathbf{P}_{\tau \sim \pi_{\text{off}}}\left(|f(\tau_{1:h-1}) - r| \le \eta \mid (s_h, a_h, s_{h+1}) \in \tau\right) = \mathbf{P}_{\tau \sim \pi_{\text{off}}}\left(|f(\tau_{1:h-1}) - r| \le \eta \mid s_h \in \tau\right) \ge 1 - p$$

and

$$\mathbf{P}_{\tau \sim \pi_{\text{off}}}\left(|f(\tau_{h:H}) + r'| \le \eta \mid (s_h, a_h, s_{h+1}) \in \tau\right) = \mathbf{P}_{\tau \sim \pi_{\text{off}}}\left(|f(\tau_{h:H}) + r'| \le \eta \mid s_{h+1} \in \tau\right) \ge 1 - p.$$

By union bound, we obtain

$$\mathbf{P}_{\tau \sim \pi_{\text{off}}}\left(|f(\tau_{1:h-1}) - r| \le \eta \text{ and } |f(\tau_{h:H}) + r'| \le \eta \mid (s_h, a_h, s_{h+1}) \in \tau\right) \ge 1 - 2p.$$

Additionally $(s_h, a_h, s_{h+1}) \in \mathcal{U}'_h$ implies $|f(s_h, a_h) + r - r'| \ge 3\eta$. Therefore, for any $(s_h, a_h, s_{h+1}) \in \mathcal{U}'_h$,

$$\mathbf{P}_{\tau \sim \pi_{\text{off}}}\left(|f(\tau)| \ge \eta \mid (s_h, a_h, s_{h+1}) \in \tau\right) \ge 1 - 2p.$$

This leads to

$$L(\eta) = \mathbf{P}_{\tau \sim \pi_{\text{off}}}(|f(\tau)| \ge \eta) \ge \mathbf{P}_{\tau \sim \pi_{\text{off}}}(\tau \cap \mathcal{U}'_h \ne \emptyset) \cdot (1 - 2p),$$

and Eq. (18) follows. □

Combining Eq. (18) and the two inequalities in Eq. (17), and in view of Eq. (16), we obtain

$$\mathbf{P}_{\tau \sim \pi_{\text{off}}}(\tau \cap \mathcal{U}_h = \emptyset) \le \frac{2L(2\eta/3)}{1-p} + \frac{8L(\eta/3)}{p} + \frac{L(\eta)}{1-2p}.$$

Next notice from the definition of state-action concentrability, we have for any policy $\pi$ and layer $h \in [H]$,

$$\sup_{s_h \in \mathcal{S}_h, a_h \in \mathcal{A}, s_{h+1} \in \mathcal{S}_{h+1}} \frac{d^\pi(s_h, a_h, s_{h+1})}{d^{\pi_{\text{off}}}(s_h, a_h, s_{h+1})}$$

$$= \sup_{s_h \in \mathcal{S}_h, a_h \in \mathcal{A}, s_{h+1} \in \mathcal{S}_{h+1}} \frac{d^\pi(s_h, a_h) \cdot T(s_{h+1} \mid s_h, a_h)}{d^{\pi_{\text{off}}}(s_h, a_h) \cdot T(s_{h+1} \mid s_h, a_h)}$$

$$= \sup_{s_h \in \mathcal{S}_h, a_h \in \mathcal{A}} \frac{d^\pi(s,a)}{d^{\pi_{\text{off}}}(s,a)} \le C_{\text{sa}}(\pi, \pi_{\text{off}}),$$

which implies that for any policy $\pi$ and layer $h \in [H]$,

$$\mathbf{P}_{\tau \sim \pi}(\tau \cap \mathcal{U}_h = \emptyset) = \sum_{(s_h, a_h, s_{h+1}) \in (\mathcal{S}_h \times \mathcal{A} \times \mathcal{S}_{h+1}) \setminus \mathcal{U}_h} d^\pi(s_h, a_h, s_{h+1})$$

$$\le C_{\text{sa}}(\pi, \pi_{\text{off}}) \cdot \sum_{(s_h, a_h, s_{h+1}) \in (\mathcal{S}_h \times \mathcal{A} \times \mathcal{S}_{h+1}) \setminus \mathcal{U}_h} d^{\pi_{\text{off}}}(s_h, a_h, s_{h+1})$$

$$= C_{\text{sa}}(\pi, \pi_{\text{off}}) \cdot \mathbf{P}_{\tau \sim \pi_{\text{off}}}(\tau \cap \mathcal{U}_h = \emptyset)$$

$$\le C_{\text{sa}}(\pi, \pi_{\text{off}}) \cdot \left( \frac{2L(2\eta/3)}{1-p} + \frac{8L(\eta/3)}{p} + \frac{L(\eta)}{1-2p} \right).$$

Hence by union bound, we have for any policy $\pi$,

$$\mathbf{P}_{\tau \sim \pi}(\tau \cap \mathcal{U}_h \ne \emptyset, \ \forall h \in [H]) \ge 1 - H C_{\text{sa}}(\pi, \pi_{\text{off}}) \cdot \left( \frac{2L(2\eta/3)}{1-p} + \frac{8L(\eta/3)}{p} + \frac{L(\eta)}{1-2p} \right).$$

Finally, we have the last claim showing that if for all $h \in [H]$, $(s_h, a_h, s_{h+1}) \in \mathcal{U}_h$, then the total reward of the trajectory can be upper bounded.

**Claim 4.** For any trajectory $\tau = (s_1, a_1, \cdots, s_H, a_H)$ where $(s_h, a_h, s_{h+1}) \in \mathcal{U}_h$ for every $h$, we have

$$|f(\tau)| \le 5H\eta.$$

**Proof.** We define tuple $u_h = (s_h, a_h, s_{h+1})$ for $h \in [H]$. According to the definition of $\mathcal{U}_h$, there exist real numbers $r(u_h) \in \mathbb{R}$ and $r'(u_{h+1}) \in \mathbb{R}$ such that for any $h \in [H]$,

$$s_h \in \mathcal{S}_h(r(u_h), \eta, p), \quad s_{h+1} \in \mathcal{S}_h(r'(u_h), \eta, p), \quad \text{and} \quad |f(s_h, a_h) + r(u_h) - r'(u_h)| \le 3\eta.$$

Compare the condition on $s_h \in \mathcal{S}_h(r(u_h), \eta, p)$ with $s_h \in \mathcal{S}_h(r'(u_{h-1}), \eta, p)$, we have

$$\mathbf{P}_{\tau \sim \pi_{\text{off}}}\left( |r(u_h) - f(\tau_{1:h-1})| \le \eta \ \Big| \ s_h \in \tau \right) \ge 1 - p \quad \text{and} \quad \mathbf{P}_{\tau \sim \pi_{\text{off}}}\left( |r'(u_{h-1}) - f(\tau_{1:h-1})| \le \eta \ \Big| \ s_h \in \tau \right) \ge 1 - p.$$

When $p < 1/2$, by union bound we obtain

$$\mathbf{P}_{\tau \sim \pi_{\text{off}}}\left( |r(u_h) - f(\tau_{1:h-1})| \le \eta \text{ and } |r'(u_{h-1}) - f(\tau_{1:h-1})| \le \eta \ \Big| \ s_h \in \tau \right) \ge 1 - 2p > 0,$$

which implies that there exists a trajectory $\tau$ such that $|r(u_h) - f(\tau_{1:h-1})| \le \eta$ and $|r'(u_{h-1}) - f(\tau_{1:h-1})| \le \eta$ both hold. Hence we obtain

$$|r(u_h) - r'(u_{h-1})| \le |r(u_h) - f(\tau_{1:h-1})| + |f(\tau_{1:h-1}) - r'(u_{h-1})| \le \eta + \eta = 2\eta.$$

Therefore, we obtain that

$$|f(\tau)| = \left| \sum_{h=1}^H f(s_h, a_h) \right| \le 3H\eta + \left| \sum_{h=1}^H \{r'(u_h) - r(u_h)\} \right| \le 5H\eta + |r'(u_H) - r(u_1)| = 5H\eta,$$

where the last equality uses the fact that $s_{H+1}$ is the notational terminal state and $s_1$ is in the first layer. $\qquad \square$

According to the previous proofs, we obtain that

$$\mathbf{P}_{\tau \sim \pi}\left( |f(\tau)| \ge 5H\eta \right) \le H C_{\text{sa}}(\pi, \pi_{\text{off}}) \cdot \left( \frac{2L(2\eta/3)}{1-p} + \frac{8L(\eta/3)}{p} + \frac{L(\eta)}{1-2p} \right).$$

By choosing $p = 1/3$ and replacing $\eta$ by $\eta/(5H)$, we have

$$\mathbf{P}_{\tau \sim \pi} \left( |f(\tau)| \geq \eta \right) \leq H C_{\mathsf{sa}}(\pi, \pi_{\mathrm{off}}) \cdot \left( 6L(2\eta/(15H)) + 24L(\eta/(15H)) + 3L(\eta/(5H)) \right)$$

$$\overset{(i)}{\leq} 33 H C_{\mathsf{sa}}(\pi, \pi_{\mathrm{off}}) \cdot L(\eta/(15H)) = 33 H C_{\mathsf{sa}}(\pi, \pi_{\mathrm{off}}) \cdot \mathbf{P}_{\tau \sim \pi_{\mathrm{off}}} \left( |f(\tau)| \geq \eta/(15H) \right),$$

where $(i)$ uses the fact that $L(\eta)$ is monotonically decreasing with $\eta$. Hence we obtain

$$\mathbb{E}_{\tau \sim \pi} \left[ f(\tau)^2 \right] \overset{(i)}{=} \int_0^1 2\eta \cdot \mathbf{P}_{\tau \sim \pi} \left( |f(\tau)| \geq \eta \right) d\eta$$

$$\leq \int_0^1 2\eta \cdot 33 H C_{\mathsf{sa}}(\pi, \pi_{\mathrm{off}}) \cdot \mathbf{P}_{\tau \sim \pi_{\mathrm{off}}} \left( |f(\tau)| \geq \eta/(15H) \right) d\eta$$

$$\leq \int_0^{15H} 2\eta \cdot 33 H C_{\mathsf{sa}}(\pi, \pi_{\mathrm{off}}) \cdot \mathbf{P}_{\tau \sim \pi_{\mathrm{off}}} \left( |f(\tau)| \geq \eta/(15H) \right) d\eta$$

$$= 7425 H^3 C_{\mathsf{sa}}(\pi, \pi_{\mathrm{off}}) \cdot \int_0^1 2(\eta/(15H)) \cdot \mathbf{P}_{\tau \sim \pi_{\mathrm{off}}} \left( |f(\tau)| \geq \eta/(15H) \right) d(\eta/(15H))$$

$$= 7425 H^3 C_{\mathsf{sa}}(\pi, \pi_{\mathrm{off}}) \cdot \mathbb{E}_{\tau \sim \pi_{\mathrm{off}}} [f(\tau)^2],$$

where $(i)$ uses integration by parts and also the assumption that for any trajectory $\tau$, $f(\tau) \in [-1, 1]$. Additionally, we upper bound the expected total reward under policy $\pi$ with Cauchy-Schwarz inequality:

$$\mathbb{E}_{\tau \sim \pi}[|f(\tau)|] \leq \sqrt{\mathbb{E}_{\tau \sim \pi}[f(\tau)^2]} \lesssim \sqrt{H^3 C_{\mathsf{sa}}(\pi, \pi_{\mathrm{off}}) \cdot \mathbb{E}_{\tau \sim \pi_{\mathrm{off}}}[f(\tau)^2]}.$$

$\square$

### B.2. Missing Proofs in Section 3.1

**Proof of Theorem 1.** For any reward model $r$, we use $r^\star[r] = r - r^\star$ to denote the difference between reward model $r$ and $r^\star$. Then we have

$$J_{r^\star[r]}(\pi) = J_r(\pi) - J(\pi),$$

For any reward $r \in \mathcal{R}$, we also have

$$\sum_{\tau \in \mathcal{D}_{\mathrm{orm}}} \left( r(\tau) - r^\star(\tau) \right)^2 = \sum_{\tau \in \mathcal{D}_{\mathrm{orm}}} \left( r^\star[r](\tau) \right)^2.$$

We notice that according to our assumption on the MDPs, for any trajectory $\tau$ and reward model $r$, $r(\tau) \in [0, 1]$, hence $r^\star[r](\tau) \in [-1, 1]$. According to Foster et al. (2021, Lemma A.3) and union bound over $\mathcal{R}$, with probability $1 - p$ we have for any $r \in \mathcal{R}$,

$$\left| \frac{1}{|\mathcal{D}_{\mathrm{orm}}|} \sum_{\tau \in \mathcal{D}_{\mathrm{orm}}} (r^\star[r](\tau))^2 - \mathbb{E}_{\tau \sim \pi_{\mathrm{off}}} \left[ (r^\star[r](\tau))^2 \right] \right| \leq \frac{1}{2} \cdot \mathbb{E}_{\tau \sim \pi_{\mathrm{off}}} \left[ (r^\star[r](\tau))^2 \right] + \frac{4 \log(2|\mathcal{R}|/p)}{|\mathcal{D}_{\mathrm{orm}}|}. \quad (19)$$

When Eq. (19) holds for all $r \in \mathcal{R}$, since $\widehat{r}$ is the solution of optimization problem Eq. (3), we have

$$\mathbb{E}_{\tau \sim \pi_{\mathrm{off}}} \left[ (r^\star[\widehat{r}](\tau))^2 \right] \leq \frac{2}{|\mathcal{D}_{\mathrm{orm}}|} \sum_{\tau \in \mathcal{D}_{\mathrm{orm}}} (r^\star[\widehat{r}](\tau))^2 + \frac{8 \log(2|\mathcal{R}|/p)}{|\mathcal{D}_{\mathrm{orm}}|}$$

$$\leq \frac{2}{|\mathcal{D}_{\mathrm{orm}}|} \sum_{\tau \in \mathcal{D}_{\mathrm{orm}}} (r^\star[r^\star](\tau))^2 + \frac{8 \log(2|\mathcal{R}|/p)}{|\mathcal{D}_{\mathrm{orm}}|}$$

$$\leq 3 \mathbb{E}_{\tau \sim \pi_{\mathrm{off}}} \left[ (r^\star[r^\star](\tau))^2 \right] + \frac{16 \log(2|\mathcal{R}|/p)}{|\mathcal{D}_{\mathrm{orm}}|}$$

$$= \frac{16 \log(2|\mathcal{R}|/p)}{|\mathcal{D}_{\mathrm{orm}}|},$$

where the last equation uses $r^\star[r^\star] = r^\star - r^\star = 0$. Therefore, according to Lemma 3, with probability at least $1 - p$ we have for any policy $\pi$,

$$|J_{\widehat{r}}(\pi) - J(\pi)| = |J_{r^\star[\widehat{r}]}(\pi)| \lesssim \sqrt{H^3 \cdot C_{\mathsf{sa}}(\pi, \pi_{\mathsf{off}}) \cdot \mathbb{E}_{\tau \sim \pi_{\mathsf{off}}}\left[(r^\star[\widehat{r}](\tau))^2\right]} \lesssim H^{3/2} \cdot \sqrt{\frac{C_{\mathsf{sa}}(\pi, \pi_{\mathsf{off}}) \cdot \log(|\mathcal{R}|/p)}{|\mathcal{D}_{\mathsf{orm}}|}}.$$

$\square$

**Proof of Corollary 2.** Suppose $\pi^\star$ to be the best policy of the true MDP $M^\star$. By Theorem 1, with probability at least $1 - \delta$, for any policy $\pi$ we have

$$|J(\pi) - J_{\widehat{r}}(\pi)| \lesssim \sqrt{\frac{H^3 C_{\mathsf{sa}}(\pi, \pi_{\mathsf{off}}) \cdot \log(|\mathcal{R}|/\delta)}{|\mathcal{D}_1|}}.$$

Especially since $\pi^\star, \widehat{\pi} \in \Pi$, we have

$$|J(\pi^\star) - J_{\widehat{r}}(\pi^\star)| \lesssim \sqrt{\frac{H^3 C_{\mathsf{sa}}(\pi^\star, \pi_{\mathsf{off}}) \cdot \log(|\mathcal{R}|/\delta)}{|\mathcal{D}_1|}} \leq \sqrt{\frac{H^3 \sup_{\pi \in \Pi} C_{\mathsf{sa}}(\pi^\star, \pi_{\mathsf{off}}) \cdot \log(|\mathcal{R}|/\delta)}{|\mathcal{D}_1|}}$$

and also

$$|J(\widehat{\pi}) - J_{\widehat{r}}(\widehat{\pi})| \lesssim \sqrt{\frac{H^3 C_{\mathsf{sa}}(\widehat{\pi}, \pi_{\mathsf{off}}) \cdot \log(|\mathcal{R}|/\delta)}{|\mathcal{D}_1|}} \leq \sqrt{\frac{H^3 \sup_{\pi \in \Pi} C_{\mathsf{sa}}(\pi^\star, \pi_{\mathsf{off}}) \cdot \log(|\mathcal{R}|/\delta)}{|\mathcal{D}_1|}}.$$

Next, by calling algorithm $\mathfrak{A}$, with probability at least $1 - \delta$ we have

$$\max_\pi J_{\widehat{r}}(\pi) - J_{\widehat{r}}(\widehat{\pi}) \leq \varepsilon_{\mathsf{alg}}(|\mathcal{D}|_2, \delta),$$

Hence with probability at least $1 - 2\delta$,

$$\max_\pi J(\pi) - J(\widehat{\pi}) = J(\pi^\star) - J(\widehat{\pi})$$

$$\leq J_{\widehat{r}}(\pi^\star) - J_{\widehat{r}}(\widehat{\pi}) + \mathcal{O}\left(\sqrt{\frac{H^3 C_{\mathsf{sa}}(\Pi, \pi_{\mathsf{off}}) \cdot \log(|\mathcal{M}|/\delta)}{|\mathcal{D}_1|}}\right)$$

$$\leq \max_\pi J_{\widehat{r}}(\pi) - J_{\widehat{r}}(\widehat{\pi}) + \mathcal{O}\left(\sqrt{\frac{H^3 C_{\mathsf{sa}}(\Pi, \pi_{\mathsf{off}}) \cdot \log(|\mathcal{M}|/\delta)}{|\mathcal{D}_1|}}\right)$$

$$\lesssim \varepsilon_{\mathsf{alg}}(|\mathcal{D}|_2, \delta) + \sqrt{\frac{H^3 C_{\mathsf{sa}}(\Pi, \pi_{\mathsf{off}}) \cdot \log(|\mathcal{M}|/\delta)}{|\mathcal{D}_1|}}.$$

$\square$

## B.3. Missing Proofs in Section 3.4

In the following, we will prove Theorem 4 and Theorem 5. Before the proofs, we first present several useful lemmas.

**Lemma 8** (Lemma A.1 in Song et al. 2024). *Suppose Assumption 1 holds, and $\widehat{\pi}$ satisfies*

$$\widehat{\pi} = \arg\max_{\pi \in \Pi} \mathbb{E}_{\tau \sim \pi}\left[\widehat{r}(\tau) - \log \frac{\pi(\tau)}{\pi_{\mathsf{ref}}(\tau)}\right] \tag{20}$$

*then we have*

$$\mathcal{J}_\beta(\pi_\beta^\star) - \mathcal{J}_\beta(\widehat{\pi}) \leq \mathbb{E}_{\tau \sim \pi_\beta^\star, \tilde{\tau} \sim \widehat{\pi}}\left[r^\star(\tau) - r^\star(\tilde{\tau}) - \widehat{r}(\tau) + \widehat{r}(\tilde{\tau})\right].$$

**Proof of Lemma 8.** We calculate

$$\mathcal{J}_\beta(\pi_\beta^\star) - \mathcal{J}_\beta(\widehat{\pi}) = \mathbb{E}_{\tau \sim \pi_\beta^\star}\left[r^\star(\tau) - \log \frac{\pi_\beta^\star(\tau)}{\pi_{\mathsf{ref}}(\tau)}\right] - \mathbb{E}_{\tau \sim \widehat{\pi}}\left[r^\star(\tau) - \log \frac{\widehat{\pi}(\tau)}{\pi_{\mathsf{ref}}(\tau)}\right]$$

$$
= \mathbb{E}_{\tau \sim \pi_{\beta}^{\star}} \left[ \widehat{r}(\tau) - \log \frac{\pi_{\beta}^{\star}(\tau)}{\pi_{\mathsf{ref}}(\tau)} \right] - \mathbb{E}_{\tau \sim \widehat{\pi}} \left[ \widehat{r}(\tau) - \log \frac{\widehat{\pi}(\tau)}{\pi_{\mathsf{ref}}(\tau)} \right]
$$

$$
+ \mathbb{E}_{\tau \sim \pi_{\beta}^{\star}} [r^{\star}(\tau) - \widehat{r}(\tau)] - \mathbb{E}_{\tau \sim \widehat{\pi}} [r^{\star}(\tau) - \widehat{r}(\tau)]
$$

$$
\overset{(i)}{\leq} \mathbb{E}_{\tau \sim \pi_{\beta}^{\star}} [r^{\star}(\tau) - \widehat{r}(\tau)] - \mathbb{E}_{\tau \sim \widehat{\pi}} [r^{\star}(\tau) - \widehat{r}(\tau)]
$$

$$
= \mathbb{E}_{\tau \sim \pi_{\beta}^{\star}, \tilde{\tau} \sim \widehat{\pi}} [r^{\star}(\tau) - r^{\star}(\tilde{\tau}) - \widehat{r}(\tau) + \widehat{r}(\tilde{\tau})],
$$

where in $(i)$ we uses Eq. (20) and the realizability assumption Assumption 1. $\qquad \square$

**Lemma 9** (Lemma C.5 in Xie et al. 2024). *We define reward model $\widehat{r}$:*

$$
\widehat{r}(\tau) = \log \frac{\widehat{\pi}(\tau)}{\pi_{\mathsf{ref}}(\tau)},
$$

*where $\widehat{\pi}$ is given in Eq. (9). Then with probability at least $1 - \delta$, we have*

$$
\mathbb{E}_{\tau, \tilde{\tau} \overset{\mathrm{iid}}{\sim} \pi_{\mathsf{ref}}} \left[ (r^{\star}(\tau) - r^{\star}(\tilde{\tau}) - \widehat{r}(\tau) + \widehat{r}(\tilde{\tau}))^2 \right] \leq \kappa^2 \cdot \frac{2 \log(|\Pi|/\delta)}{|\mathcal{D}|},
$$

*where $\kappa = 16 V_{\max} e^{2 V_{\max}}$.*

**Lemma 10.** *For MDP $M = (\mathcal{S}, \mathcal{A}, T, r, H)$ with reward function $f : \mathcal{S} \times \mathcal{A} \to [-1, 1]$, then for any policy $\pi$ and $\tilde{\pi}$, we have*

$$
\mathbb{E}_{\tau \sim \pi, \tilde{\tau} \sim \tilde{\pi}} [|f(\tau) - f(\tilde{\tau})|] \lesssim \sqrt{H^3 (C_{\mathsf{sa}}(\pi, \pi_{\mathsf{off}}) \vee C_{\mathsf{sa}}(\tilde{\pi}, \pi_{\mathsf{off}})) \cdot \mathbb{E}_{\tau, \tilde{\tau} \overset{\mathrm{iid}}{\sim} \pi_{\mathsf{off}}} \left[ (f(\tau) - f(\tilde{\tau}))^2 \right]},
$$

*where $C_{\mathsf{sa}}(\pi, \pi_{\mathsf{off}})$ is defined in Eq. (5).*

**Proof of Lemma 10.** Suppose the layered state space representation is $\mathcal{S} = \cup_{h=1}^{H} \mathcal{S}_h$. We construct a new MDP $M'$ with horizon $2H$. The state spaces of $M'$ among layer $1$ to $H$, and among layer $H + 1$ to $2H$, are both $\mathcal{S}_1, \cdots, \mathcal{S}_H$. The action space of $\mathcal{M}'$ is $\mathcal{A}$. The transition model of $M'$ follows transition model $T$ in the first $H$ layers, and then transits to the initial state $s_1$ of $M$ and follows transition model $T$ again in the last $H$ layers as well. The reward model $g$ in the first $H$ layers are set to be $f$ and in the last $H$ layers are set to be $-f$.

We define the policy $\pi'$ of MDP $M'$, which follows policy $\pi$ in the first $H$ layers, and follows policy $\tilde{\pi}$ in the last $H$ layers. We further define the policy $\pi'_{\mathsf{off}}$ of MDP $M'$, which follows policy $\pi_{\mathsf{off}}$ in the first $H$ layers, and in the last $H$ layers as well. Then it is easy to see that

$$
\mathbb{E}_{\tau \sim \pi, \tilde{\tau} \sim \tilde{\pi}} [|f(\tau) - f(\tilde{\tau})|] = \mathbb{E}_{\tau' \sim \pi'} [|g(\tau')|],
$$

where the right hand side is the expected rewards in the new MDP $M'$. We also have

$$
\mathbb{E}_{\tau, \tilde{\tau} \overset{\mathrm{iid}}{\sim} \pi_{\mathsf{off}}} \left[ \left( f(\tau) - f(\tilde{\tau}))^2 \right)^2 \right] = \mathbb{E}_{\tau' \sim \pi_{\mathsf{off}}} \left[ g(\tau')^2 \right].
$$

Next, according to the construction of $M'$, we have

$$
C_{\mathsf{sa}}(\pi, \pi_{\mathsf{off}}) \vee C_{\mathsf{sa}}(\tilde{\pi}, \pi_{\mathsf{off}}) = \sup_{h \in [2H]} \sup_{s \in \mathcal{S}_h, a \in \mathcal{A}} \frac{d^{\pi'}(s, a)}{d^{\pi'_{\mathsf{off}}}(s, a)}.
$$

Hence according to Lemma 3, we have

$$
\mathbb{E}_{\tau' \sim \pi'} [|g(\tau')|] \lesssim \sqrt{H^3 \cdot \sup_{h \in [2H]} \sup_{s \in \mathcal{S}_h, a \in \mathcal{A}} \frac{d^{\pi'}(s, a)}{d^{\pi'_{\mathsf{off}}}(s, a)} \cdot \mathbb{E}_{\tau' \sim \pi_{\mathsf{off}}} [g(\tau')^2]}
$$

$$
= \sqrt{H^3 (C_{\mathsf{sa}}(\pi, \pi_{\mathsf{off}}) \vee C_{\mathsf{sa}}(\tilde{\pi}, \pi_{\mathsf{off}})) \cdot \mathbb{E}_{\tau' \sim \pi_{\mathsf{off}}} [g(\tau')^2]},
$$

which implies that

$$
\mathbb{E}_{\tau \sim \pi, \tilde{\tau} \sim \tilde{\pi}} [|f(\tau) - f(\tilde{\tau})|] \lesssim \sqrt{H^3 (C_{\mathsf{sa}}(\pi, \pi_{\mathsf{off}}) \vee C_{\mathsf{sa}}(\tilde{\pi}, \pi_{\mathsf{off}})) \cdot \mathbb{E}_{\tau, \tilde{\tau} \overset{\mathrm{iid}}{\sim} \pi_{\mathsf{off}}} \left[ (f(\tau) - f(\tilde{\tau}))^2 \right]}.
$$

$\qquad \square$

Now we are ready to prove Theorem 4 and Theorem 5.

***Proof of Theorem 4.*** According to Lemmas 9 and 10, for any policy $\pi$, with probability at least $1 - \delta$,

$$\mathbb{E}_{\tau \sim \pi, \tau' \sim \pi'}[|r^\star(\tau) - r^\star(\tau') - \widehat{r}(\tau) + \widehat{r}(\tau')|]$$

$$\lesssim \sqrt{H^3 \cdot (C_{\mathsf{sa}}(\pi, \pi_{\mathsf{ref}}) \vee C_{\mathsf{sa}}(\pi', \pi_{\mathsf{ref}})) \cdot \mathbb{E}_{\tau, \tilde{\tau} \overset{\mathrm{iid}}{\sim} \pi_{\mathsf{ref}}} \left[ (r^\star(\tau) - r^\star(\tilde{\tau}) - \widehat{r}(\tau) + \widehat{r}(\tilde{\tau}))^2 \right]}$$

$$\lesssim H^{3/2} V_{\max} e^{2V_{\max}} \cdot \sqrt{\frac{(C_{\mathsf{sa}}(\pi, \pi_{\mathsf{ref}}) \vee C_{\mathsf{sa}}(\pi', \pi_{\mathsf{ref}})) \log(|\mathcal{R}|/\delta)}{|\mathcal{D}|}},$$

where the second line uses Lemma 9 with the class of policies $\{\widehat{\pi} : \widehat{\pi}(\tau) \propto \pi_{\mathsf{ref}}(\tau) \exp(r(\tau)) \; \forall r \in \mathcal{R}\}$. $\qquad\square$

***Proof of Theorem 5.*** Let $\widehat{\pi}$ to be the policy given in Eq. (9). We define the reward model $\widehat{r}$ as

$$\widehat{r}(\tau) = \log \frac{\widehat{\pi}(\tau)}{\pi_{\mathsf{ref}}(\tau)}.$$

Then it is easy to see that $\widehat{\pi}$ is the solution of Eq. (20) according to the above reward model. Since $\widehat{\pi} \in \Pi$ and $\pi_\beta^\star \in \Pi$ according to Assumption 1, with probability at least $1 - \delta$ we have

$$\mathcal{J}_\beta(\pi_\beta^\star) - \mathcal{J}_\beta(\widehat{\pi}) \overset{(i)}{\leq} \mathbb{E}_{\tau \sim \pi_\beta^\star, \tilde{\tau} \sim \widehat{\pi}} \left[ r^\star(\tau) - r^\star(\tilde{\tau}) - \widehat{r}(\tau) + \widehat{r}(\tilde{\tau}) \right]$$

$$\overset{(ii)}{\lesssim} \sqrt{H^3 \cdot C_{\mathsf{sa}}(\Pi, \pi_{\mathsf{ref}}) \cdot \mathbb{E}_{\tau, \tilde{\tau} \overset{\mathrm{iid}}{\sim} \pi_{\mathsf{ref}}} \left[ (r^\star(\tau) - r^\star(\tilde{\tau}) - \widehat{r}(\tau) + \widehat{r}(\tilde{\tau}))^2 \right]}$$

$$\overset{(iii)}{\lesssim} H^{3/2} V_{\max} e^{2V_{\max}} \cdot \sqrt{\frac{C_{\mathsf{sa}}(\Pi, \pi_{\mathsf{ref}}) \log(|\Pi|/\delta)}{|\mathcal{D}|}},$$

where $(i)$ uses Lemma 8, $(ii)$ uses Lemma 10 with the reward model to be $r - \widehat{r}$, and $(iii)$ uses Lemma 9. $\qquad\square$

## C. Analysis of ARMOR with Total Reward

ARMOR is an offline RL algorithm introduced by Xie et al. (2022a). Given $n$ trajectories of data in the form of $(s_1, a_1, r_1, \cdots, s_H, a_H, r_H) \sim \pi_{\mathrm{off}}$, they proved that for the optimal policy $\pi^\star$, the output policy $\widehat{\pi}$ of their algorithm satisfies

$$J(\pi^\star) - J(\widehat{\pi}) \lesssim \sqrt{\frac{H^2 \cdot C(\pi^\star, \pi_{\mathrm{off}}) \log(|\mathcal{M}|/\delta)}{n}}. \tag{21}$$

Notably, this error bound scales with the best-policy concentrability $C(\pi^\star, \pi_{\mathrm{off}})$ rather than the all-policy concentrability $\sup_\pi C(\pi, \pi_{\mathrm{off}})$ shown in Corollary 2. A natural question arises: can we develop a variant of ARMOR that takes data in the form of trajectory plus total reward, i.e., $(s_1, a_1, \cdots, s_H, a_H, R)$, while maintaining the sample complexity that scales with best-policy concentrability instead of all-policy concentrability? In this section, we answer this question affirmatively by presenting and analyzing such a variant of the ARMOR algorithm.

Suppose the learner is given a model class $\mathcal{M}$, which realizes the ground truth model $M^\star$. The learner also have access to some offline batched data $\mathcal{D}$, which is collected from policy $\pi_{\mathrm{off}}$. Specifically, $\mathcal{D}$ consists of i.i.d. sampled trajectories $\tau = (s_1, a_1, \cdots, s_H, a_H)$ together with its total reward $R = r(\tau)$. Each of the trajectories is collected by executing $\pi_{\mathrm{off}}$ under the ground truth MDP $M^\star$.

We consider Algorithm 2, which is a variant of the ARMOR algorithm introduced in Xie et al. (2022a) after specifically taylored for data of trajectories with total reward. We have the following guarantee on its sample complexity, which only relies on the single policy concentrability $C_{\mathsf{sa}}(\pi^\star, \pi_{\mathrm{off}})$.

**Theorem 11.** *Suppose the model class $\mathcal{M}$ realizes the ground truth model $M^\star$. Then there exists some positive constant $c$ such that for any $\delta > 0$, by letting $\alpha = c \cdot \log(|\mathcal{M}|/\delta)$, with probability at least $1 - \delta$, the output of Algorithm 2 satisfies*

$$J_{M^\star}(\pi^\star) - J_{M^\star}(\widehat{\pi}) \lesssim \sqrt{\frac{H^3 \cdot C_{\mathsf{sa}}(\pi^\star, \pi_{\mathrm{off}}) \log(|\mathcal{M}|/\delta)}{n}}$$

---

**Algorithm 2** ARMOR with Total Rewards

1: **Input:** Batch data $\mathcal{D}$, model class $\mathcal{M}$, parameter $\alpha$.
2: Construct version space

$$\mathcal{M}_\alpha = \left\{ M \in \mathcal{M} : \max_{M' \in \mathcal{M}} \mathcal{L}_\mathcal{D}(M') - \mathcal{L}_\mathcal{D}(M) \le \alpha \right\},$$

where for MDP model $M$ with transition model $P_M$ and reward function $r_M$,

$$\mathcal{L}_\mathcal{D}(M) \coloneqq \sum_{(s_1, a_1, \cdots, s_H, a_H, R) \in \mathcal{D}} \left[ \sum_{h=1}^{H-1} \log P_M(s_{h+1} \mid s_h, a_h) - \left( \sum_{h=1}^{H} r_M(s_h, a_h) - R \right)^2 \right]$$

3: Output the best policy with pessimism:

$$\widehat{\pi} = \arg\max_\pi \min_{M \in \mathcal{M}_\alpha} J_M(\pi),$$

where $J_M(\pi)$ denotes the value function of policy $\pi$ under MDP $M$

---

***Proof of Theorem 11.*** Adopting the similar way as Xie et al. (2022a), we let

$$\ell_\mathcal{D}(M) = \prod_{(s_1, a_1, \cdots, s_H, a_H, R) \in \mathcal{D}} \prod_{h=1}^{H-1} P_M(s_{h+1}, s_h, a_h).$$

Then according to Xie et al. (2022a, Lemma 8), with probability at least $1 - \delta$, we have

$$\max_{M \in \mathcal{M}} \log \ell_\mathcal{D}(M) - \ell_\mathcal{D}(M^\star) \le \log\left( \frac{|\mathcal{M}|}{\delta} \right),$$

where $M^\star$ denotes the ground truth model. Additionally, according to Xie et al. (2021a, Theorem A.1) (by letting $\gamma = 0$ and merge states or actions across the entire horizon into one state or action), we have

$$\sum_{(\tau, R) \in \mathcal{D}} (r_{M^\star}(\tau) - R)^2 - \min_{M \in \mathcal{M}} \sum_{(\tau, R) \in \mathcal{D}} (r_M(\tau) - R)^2 \lesssim \log\left( \frac{|\mathcal{M}|}{\delta} \right).$$

Combining the above inequalities together, we obtain that

$$\max_{M \in \mathcal{M}} \mathcal{L}_\mathcal{D}(M) - \mathcal{L}_\mathcal{D}(M^\star) \lesssim \log\left( \frac{|\mathcal{M}|}{\delta} \right).$$

Hence with our choice of $\alpha$, with probability at least $1 - \delta$ we have $M^\star \in \mathcal{M}$. Next, by Xie et al. (2022a, Lemma 7), with probability at least $1 - \delta$, for any $M \in \mathcal{M}$,

$$\mathbb{E}_{(s_1, a_1, \cdots, s_H, a_H) \sim \pi_{\text{off}}} \left[ \sum_{h=1}^{H-1} D_{\text{TV}}(P_M(s_{h+1} \mid s_h, a_h), P_{M^\star}(s_{h+1} \mid s_h, a_h)) + \left( \sum_{h=1}^{H} r_M(s_h, a_h) - r_{M^\star}(s_h, a_h) \right)^2 \right]$$

$$\lesssim \frac{\max_{M' \in \mathcal{M}} \mathcal{L}_\mathcal{D}(M') - \mathcal{L}_\mathcal{D}(M^\star) + \log(|\mathcal{M}|/\delta)}{n} \lesssim \frac{\log(|\mathcal{M}|/\delta)}{n}. \tag{22}$$

Finally, according to the optimality of $\widehat{\pi}$, if letting $\widehat{M} = \arg\min_{M \in \mathcal{M}} J_M(\widehat{\pi})$, we have

$$J_{M^\star}(\pi^\star) - J_{M^\star}(\widehat{\pi}) \le J_{M^\star}(\pi^\star) - \min_{M \in \mathcal{M}} J_M(\widehat{\pi})$$

$$\overset{(i)}{=} J_{M^\star}(\pi^\star) - \max_\pi \min_{M \in \mathcal{M}} J_M(\widehat{\pi})$$

$$\le \max_{M \in \mathcal{M}} \left\{ J_{M^\star}(\pi^\star) - J_M(\pi^\star) \right\},$$

where $(i)$ uses the definition of $\widehat{\pi}$ in Algorithm 2. According to simulation lemma (e.g., Uehara and Sun, 2021, Lemma 7), we have

$$
\begin{aligned}
|J_{M^\star}(\pi^\star) - J_M(\pi^\star)| &\leq \sum_{h=1}^{H-1} \mathbb{E}_{(s_h,a_h)\sim d^{\pi^\star}}[D_{\mathrm{TV}}(P_M(s_{h+1}\mid s_h,a_h), P_{M^\star}(s_{h+1}\mid s_h,a_h))] \\
&\quad + \mathbb{E}_{\tau\sim\pi^\star}[|r_{M^\star}(\tau) - r_M(\tau)|] \\
&\leq H^{1/2}\cdot\sqrt{C_{\mathsf{sa}}(\pi^\star,\pi_{\mathrm{off}})\sum_{h=1}^{H-1}\mathbb{E}_{(s_h,a_h)\sim d^{\pi_{\mathrm{off}}}}[D_{\mathrm{TV}}(P_M(s_{h+1}\mid s_h,a_h), P_{M^\star}(s_{h+1}\mid s_h,a_h))^2]} \\
&\quad + \mathbb{E}_{\tau\sim\pi^\star}[|r_{M^\star}(\tau) - r_M(\tau)|],
\end{aligned}
$$

where the last inequality uses the Cauchy-Schwarz inequality and the definition of state-action concentrability. Additionally, according to Lemma 3, we have

$$
\mathbb{E}_{\tau\sim\pi^\star}[|r_{M^\star}(\tau) - r_M(\tau)|] \lesssim H^{3/2}\sqrt{C_{\mathsf{sa}}(\pi^\star,\pi_{\mathrm{off}})\cdot\mathbb{E}_{\tau\sim\pi_{\mathrm{off}}}[(r_{M^\star}(\tau) - r_M(\tau))^2]}.
$$

Therefore, with probability at least $1-\delta$,

$$
\begin{aligned}
J_{M^\star}(\pi^\star) - J_{M^\star}(\widehat{\pi}) &\leq \max_{M\in\mathcal{M}}\{|J_{M^\star}(\pi^\star) - J_M(\pi^\star)|\} \lesssim \sqrt{H^3 C_{\mathsf{sa}}(\pi^\star,\pi_{\mathrm{off}})} \\
&\quad\cdot\sqrt{\mathbb{E}_{(s_1,a_1,\cdots,s_H,a_H)\sim\pi_{\mathrm{off}}}\left[\sum_{h=1}^{H-1}D_{\mathrm{TV}}(P_M(s_{h+1}\mid s_h,a_h), P_{M^\star}(s_{h+1}\mid s_h,a_h))\right] + \mathbb{E}_{\tau\sim\pi_{\mathrm{off}}}(r_M(\tau) - r_{M^\star}(\tau))^2} \\
&\lesssim \sqrt{\frac{H^3 C_{\mathsf{sa}}(\pi^\star,\pi_{\mathrm{off}})\log(|\mathcal{M}|/\delta)}{n}}
\end{aligned}
$$

where the last inequality is due to Eq. (22). $\qquad\square$

We observe that similar to the ARMOR algorithm, the sample complexity of Algorithm 2 scales with the single-policy concentrability $C_{\mathsf{sa}}(\pi^\star,\pi_{\mathrm{off}})$ rather than the all-policy concentrability $\sup_\pi C_{\mathsf{sa}}(\pi,\pi_{\mathrm{off}})$ as shown in Theorem 11. However, unlike ARMOR, Theorem 11 requires the assumption that the total reward is bounded by $[0,1]$. Without this assumption, if we include the scale of total reward in the proof of Theorem 11, we would directly obtain a sample complexity of $\sqrt{\frac{H^5\cdot C_{\mathsf{sa}}(\pi^\star,\pi_{\mathrm{off}})\log(|\mathcal{M}|/\delta)}{n}}$. Comparing this with ARMOR's sample complexity (Eq. (21)), we observe a gap of $H^3$ (when obtaining an $\varepsilon$-sub-optimality bound) in terms of the horizon dependency.

We suspect this gap arises fundamentally from the extra horizon dependency in the Change of Trajectory Measure Lemma (Lemma 3). To illustrate this, consider how changing measures affects complexity (ignoring log terms):

- Change of state-action measure: For any function $g : \mathcal{S}\times\mathcal{A}\to\mathbb{R}$,

$$
\frac{\sum_{h=1}^{H}\mathbb{E}_{(s_h,a_h)\sim d^\pi}[g(s_h,a_h)^2]}{\sum_{h=1}^{H}\mathbb{E}_{(s_h,a_h)\sim d^{\pi_{\mathrm{off}}}}[g(s_h,a_h)^2]} \leq \max_{h\in[H]}\frac{\mathbb{E}_{(s_h,a_h)\sim d^\pi}[g(s_h,a_h)^2]}{\mathbb{E}_{(s_h,a_h)\sim d^{\pi_{\mathrm{off}}}}[g(s_h,a_h)^2]} \leq \max_{h\in[H]}\sup_{s_h\in\mathcal{S}_h,a_h\in\mathcal{A}}\frac{d^\pi(s_h,a_h)}{d^{\pi_{\mathrm{off}}}(s_h,a_h)}.
$$

- Change of trajectory measure (Lemma 3): For any function $f : \mathcal{S}\times\mathcal{A}\to\mathbb{R}$,

$$
\frac{\mathbb{E}_{\tau\sim\pi}[f(\tau)^2]}{\mathbb{E}_{\tau\sim\pi_{\mathrm{off}}}[f(\tau)^2]} \lesssim H^3\max_{h\in[H]}\sup_{s_h\in\mathcal{S}_h,a\in\mathcal{A}}\frac{d^\pi(s_h,a_h)}{d^{\pi_{\mathrm{off}}}(s_h,a_h)}.
$$

While this gap in horizon dependency raises an interesting theoretical question, the precise polynomial dependence on horizon is not the main focus of our paper. We leave a more thorough investigation of this horizon dependency gap between outcome and process supervision as an interesting direction for future work.

# D. Missing Proofs in Section 4

## D.1. Proof of Theorem 6

First we present a lemma.

**Lemma 12.** *For any MDP $M = (\mathcal{S}, \mathcal{A}, P, r, H)$ and policy $\mu$, if we let $A^\mu : \mathcal{S} \times \mathcal{A} \to \mathbb{R}$ to be the advantage function of policy $\mu$ (defined in Eq. (10)), then for any policy $\pi$, we have*

$$J_{A^\mu}(\pi) = J_r(\pi) - J_r(\mu).$$

***Proof of Lemma 12.*** In view of the performance difference lemma (Kakade and Langford, 2002), we have

$$
\begin{aligned}
J_r(\pi) - J_r(\mu) &= H \cdot \mathbb{E}_{(s,a)\sim d^\pi} \left[ Q^\mu(s,a) - Q^\mu(s,\mu) \right] \\
&= H \cdot \mathbb{E}_{(s,a)\sim d^\pi} \left[ Q^\mu(s,a) - V^\mu(s) \right] \\
&= H \cdot \mathbb{E}_{(s,a)\sim d^\pi} \left[ A^\mu(s,a) \right] \\
&= J_{A^\mu}(\pi).
\end{aligned}
$$

$\square$

Next we present the proof of Theorem 6.

***Proof of Theorem 6.*** For any policy $\pi$, we decompose

$$
\begin{aligned}
J_r(\pi) - J_r(\widehat{\pi}) &= J_r(\pi) - J_{\widehat{r}}(\pi) + J_{\widehat{r}}(\pi) - J_{\widehat{r}}(\widehat{\pi}) + J_{\widehat{r}}(\widehat{\pi}) - J_r(\widehat{\pi}) \\
&\overset{(i)}{=} J_{A^\mu}(\pi) - J_{\widehat{r}}(\pi) + J_{\widehat{r}}(\pi) - J_{\widehat{r}}(\widehat{\pi}) + J_{\widehat{r}}(\widehat{\pi}) - J_{A^\mu}(\widehat{\pi}),
\end{aligned}
\tag{23}
$$

where in $(i)$ we use Lemma 12. Next, we notice that for any policy $\pi$, we can write down the value of policy $\pi$ as the inner product between the occupancy measures of policy $\pi$ and the reward function, which implies that

$$
\begin{aligned}
J_{A^\mu}(\pi) - J_{\widehat{r}}(\pi) &= H \cdot \sum_{s \in \mathcal{S}} \sum_{a \in \mathcal{A}} d^\pi(s,a) \cdot \left( A^\mu(s,a) - \widehat{r}(s,a) \right) \\
&\overset{(i)}{\leq} H \cdot \sqrt{ \sum_{s \in \mathcal{S}} \sum_{a \in \mathcal{A}} d^\pi(s,a) \cdot \left( A^\mu(s,a) - \widehat{r}(s,a) \right)^2 } \\
&\overset{(ii)}{\leq} H \cdot \sqrt{ C_{\mathsf{sa}}(\nu) \cdot \mathbb{E}_{(s,a)\sim\nu} \left[ \left( A^\mu(s,a) - \widehat{r}(s,a) \right)^2 \right] } \\
&\overset{(iii)}{\leq} H \sqrt{C_{\mathsf{sa}}(\nu)} \cdot \varepsilon_{\mathsf{stat}},
\end{aligned}
$$

where $(i)$ uses Cauchy-Schwarz inequality, $(ii)$ adopts the definition of $C_{\mathsf{sa}}(\nu)$ in Eq. (11), and finally in $(iii)$ we use Eq. (12). Additionally, according to Eq. (13) we have

$$J_{\widehat{r}}(\pi) - J_{\widehat{r}}(\widehat{\pi}) \leq \max_\pi J_{\widehat{r}}(\pi) - J_{\widehat{r}}(\widehat{\pi}) \leq \varepsilon_{\mathsf{alg}}.$$

Bringing these inequalities back to Eq. (23), we obtain that

$$J_r(\pi) - J_r(\widehat{\pi}) \leq 2H \sqrt{C_{\mathsf{sa}}(\nu)} \cdot \varepsilon_{\mathsf{stat}} + \varepsilon_{\mathsf{alg}}.$$

$\square$

## D.2. Proof of Theorem 7

***Proof of Theorem 7.*** We construct $M$ as follows: We let $H = 2$, $\mathcal{A} = \{0,1\}$, and $\mathcal{S} = \mathcal{S}_1 \cup \mathcal{S}_2$ where $\mathcal{S}_1 = \{a\}$ and $\mathcal{S}_2 = \{b, c\}$. We further define the transition model $P$ as

$$P(b \mid a, 0) = 1 \quad \text{and} \quad P(c \mid a, 1) = 1,$$

and the reward function $R$ as

$$R(a,0) = R(a,1) = 0, \quad \text{and} \quad R(b,0) = 1, R(b,1) = 0, \quad \text{and} \quad R(c,0) = \frac{2}{3}, R(c,1) = \frac{1}{2}.$$

The best policy $\pi^\star$ of this MDP $M$ satisfies

$$\pi^\star(a) = \pi^\star(b) = \pi^\star(c) = 0,$$

hence $J_r(\pi^\star) = 1$. We next choose policy $\mu$ as $\mu(a) = 0$, $\mu(b) = \mu(c) = 1$. Then we can calculate that

$$Q^\mu(b,0) = R(b,0) = 1, \quad Q^\mu(b,1) = R(b,1) = 0,$$
$$Q^\mu(c,0) = R(c,0) = \frac{2}{3}, \quad Q^\mu(c,1) = R(c,1) = \frac{1}{2},$$
$$Q^\mu(a,0) = R(a,0) + Q^\mu(b,\mu(b)) = 0, \quad Q^\mu(a,1) = R(a,1) + Q^\mu(c,\mu(c)) = \frac{1}{2}.$$

Therefore, the greedy policy $\widehat{\pi}$ with respect to the MDP with reward function to be $Q^\mu$ is

$$\widehat{\pi}(a) = 1, \quad \widehat{\pi}(b) = 0, \quad \widehat{\pi}(c) = 0,$$

which satisfies

$$\max_\pi J_r(\pi) - J_r(\widehat{\pi}) = 1 - \frac{2}{3} = \frac{1}{3}.$$

$\square$

**Remark 1.** *With the same choice of MDP $M$ and policy $\mu$ in the above proof, we can calculate the advantage function $A^\mu$ as*

$$A^\mu(b,0) = 1, \quad A^\mu(b,1) = 0, \quad \text{and} \quad A^\mu(c,0) = \frac{1}{6}, \quad A^\mu(c,1) = 0, \quad \text{and} \quad A^\mu(a,0) = 0, \quad A^\mu(a,1) = \frac{1}{2}.$$

*And it is easy to verify that the greedy policy with respect to the MDP with reward function to be $A^\mu$ coincides with $\pi^\star$.*

