# OpenReview forum: "Do We Need to Verify Step by Step? Rethinking Process Supervision from a Theoretical Perspective"
_ICML.cc/2025/Conference — ICML 2025 poster_

### Official Review · Reviewer_jQ9F · 2025-03-13

**Overall Recommendation:** 5

**Summary:**

The paper’s central takeaway is that offline RL under trajectory-wide sparse rewards (for LLMs, outcome supervision) and dense rewards (process supervision) are statistically equivalent (up to polynomial factors in the horizon).

The main contribution is the Change of Trajectory Measure Lemma, a powerful result that asymptotically bounds the quotient of the second moments for trajectory distributions from two different policies of a function on (s, a) added across the trajectory. Crucially, the bound is based on the horizon and the state-action-level distribution shift between the policies.

This Lemma is then used to show the aforementioned statistical equivalence (up to polynomial factors in the horizon) between outcome and process supervision, as well as tighter results for Preference-based RL with explicit reward modeling and DPO. Finally, the paper analyzes two common approaches to learning a process reward from outcome supervision, showing rewards that approximate the advantage function are theoretically coherent. In contrast, rewards that approximate the Q function are not consistent.

## Post–Rebuttal
I have read the concerns from other reviewers and the responses from the authors, and I maintain my original assessment.

**Claims And Evidence:**

Most of the claims made are supported by a complete and novel theoretical analysis. Nonetheless, I believe the abstract and introduction do not state clearly enough that some of the provided results (Thm 1, Cor 2, Thm 4, and Thm 5) require the assumption of a finite hypothesis set, which impacts practical applicability. Similarly, the results provided (Thm 1, Cor 2) are limited to offline RL settings, which should also be clarified more explicitly in the introduction.

**Essential References Not Discussed:**

I don’t have a specific set of missing works that should be cited, but I would propose some parts of the paper that would benefit from appropriate citations.

Proofs such as the one of Lemma 3 seem to use relatively standard ideas from the literature. I don’t want to push any specific work that inspired some of these ideas, but I believe it would be good if the authors would acknowledge the works that most influenced their approach to the proofs.

I would also appreciate relevant citations in the second column in L416 after “widely used.”

**Experimental Designs Or Analyses:**

There are no empirical results.

**Methods And Evaluation Criteria:**

Yes, claims are theoretical and accompanied by corresponding proof. Consequently, there are no empirical results, but none appear required for the paper’s soundness.

**Other Comments Or Suggestions:**

* L074, C2. The notation probably belongs more to the background than an introduction, and given that both the O notation and sigmoid are reasonably common, I would remove this section and introduce the notation the first time it appears.
* L075, C2. I would encourage using only the O notation (or the other), not both.
* L111, C2. RL is used without relating it to “Reinforcement Learning,” and “Reinforcement Learning” is probably fully spelled too many times, as well as “Large Language Model.”
* L168, C2. Here, it would be nice to clarify that you are referring to state-action rewards more explicitly, maybe through using the r notation to distinguish from R. I think this paragraph could be clearer.
* L259, C1. This paragraph could be made clearer by saying: “Indeed, many classical offline RL algorithms ... are known to achieve exactly this sample complexity under standard coverage assumptions”. Otherwise, the two sentences might seem to make different points while trying to get to the same point.
* L268, C1. Maybe it would be clearer to explicitly refer to “the single-policy concentration bound in Appendix C, ...” instead of "the data-dependent case.”
* L270, C2. "till" -> "until"
* L300, C1, and L863, among others. I would encourage using max and min instead of ∨ and ∧.
* L694, I believe there is a typo here and f(s_h) should be r(s_h).

**Other Strengths And Weaknesses:**

I believe the above conveys the strengths I find in the paper pretty comprehensively, as well as the main weaknesses (not stating adequately in the abstract and intro the offline setting and finite hypothesis set).

In terms of weaknesses, as usual, proofs could contain more text to ensure they are pleasant and amenable to the reader.

Also:
* L179, C2. To use the union bound, and of course, as is apparent from the result, a big assumption is the finiteness of |R|. This can probably be relaxed in the future, but as it currently stands, this is a big assumption that should be more clearly stated.

**Questions For Authors:**

I have no specific questions, but I welcome the author’s pointing out anything in my review that might suggest I have misunderstood anything.

**Relation To Broader Scientific Literature:**

To the best of my knowledge, the results presented in this paper are novel and very valuable. Although their positioning focuses on the timely and active field of outcome and process supervision, their result seems to extend naturally to any offline RL method with trajectory-wide sparse rewards, making the analysis broadly impactful.

More importantly, their main result, Lemma 3, is powerful and widely applicable even beyond offline RL. Their paper already shows its flexibility through the various results they derive from it, ranging from offline RL in general and outcome supervision to preference-based learning. Future works will likely benefit from and build on these results.

More specifically, position-wise, this paper is tightly relevant to recent works in LLMs where either (1) only rewards for the outcome (outcome supervision) or (2) step-by-step rewards (process supervision) are available. The community has recently been unable to reach a consensus on which of these approaches is most compelling overall, as adequately discussed in the paper. On the one hand, process supervision provides a richer signal that aids learning, while outcome supervision facilitates data collection. This paper examines the situation from a theoretical perspective, proving that outcome supervision is statistically no harder than process supervision (up to poly(H) factors), providing a solid argument to challenge what’s considered the main limitation of outcome supervision with respect to process supervision.

Regarding the framework adopted for the theory, this paper works on the “state-action coverage” assumption, which, as cited correctly in the paper, is a widely used measure of statistical difficulty in the offline setting. While this magnitude is natural for process supervision, the equivalent for outcome supervision would be the much looser trajectory coverage. Through lemma 3, this paper not only tightens results that naturally depend on trajectory-level coverage, such as therein cited reward-modeling methods (e.g., DPO), but more generally bridges an important gap in OPE (as properly cited).

Finally, the paper also provides compelling theoretical grounding for advantage-function-based “process” rewards, as done in some recent works, while shining a light on the limitations of Q-function-based rewards, as done in others. Indeed, the paper shows that the former preserves optimal policies while the latter doesn’t.

Overall, this paper significantly advances offline RL theory, with direct impact and insight on LLM post-training.

**Theoretical Claims:**

I carefully verified the proofs for the core Lemma 3 and Theorems 1 and 7. I also reviewed the proofs of the remaining results but did not reproduce them with the same level of detail. Everything seems correct.

---

> ### Author Rebuttal · Authors · 2025-04-01
>
> Thank you very much for your support.
>
> ***
>
> > Relevant Citations in L416
>
> We will add the citations accordingly.
>
> ***
> > Assumption on finiteness of $|R|$
>
> Thanks for pointing this out. Even though the current theorems are stated for finite reward models, the change-of-trajectory-measure lemma does not require the size of the reward class to be finite. We believe it is easy to generalize our results to an infinite-size parametric reward class, or even to a nonparametric reward class, with the help of classical tools for analyzing the ERM performance in parametric or nonparametric model classes. We will add an explicit discussion regarding the assumption on the finiteness of $|R|$ in our future version.
>
> ***
>
> > Typos
>
> Thanks for pointing out the typos. We will fix these in the revision.

---

### Official Review · Reviewer_yWDQ · 2025-03-14

**Overall Recommendation:** 4

**Summary:**

This paper shows a theoretical analysis of the statistical equivalence between dense reward trajectories and sparse reward trajectories, which returns the total reward at the end of the episode in offline RL.

**Post-rebuttal**
I read the responses from the authors and other reviewers.
I think this is a theory paper, as mentioned in the rebuttals, and it shows new findings.
There is a weak connection to practical algorithm designs, but this paper offers valuable insights;
difficulty in obtaining PRMs in practice, and several approaches are using outcome rewards to translate into process rewards.
advantage function is better considered than the q-function.

This paper will be more appreciated by people who are interested in theory.
Extracting more valuable insights would be even more appreciated, and providing intuition about the assumptions would also increase the impact of this work. Finally, I increased the score, reflecting the significance of the work.

**Claims And Evidence:**

The main claims are given in the form of concentration inequalities.

**Essential References Not Discussed:**

No

**Experimental Designs Or Analyses:**

NA

**Methods And Evaluation Criteria:**

NA

**Other Comments Or Suggestions:**

NA

**Other Strengths And Weaknesses:**

The strength is that the claim is very nice, if it is possible in practice. We don't need to annotate or learn a dense reward model, but sparse reward trajectories could solve all problems.
The main weakness is that it is difficult to imagine how this result can be useful.

**Questions For Authors:**

Could you provide any tangible procedure that could replace the PRM with ORM?

In practice, what will be the difficulty of applying this idea?

What assumption is not realistic?

**Relation To Broader Scientific Literature:**

This is a mathematical theory paper analyzing the error bounds in offline reinforcement learning, and I am not sure how much of the theoretical claims can actually be transferred to scientific literature.

**Theoretical Claims:**

No

---

> ### Author Rebuttal · Authors · 2025-04-01
>
> Thank you for the review. Below, we address the concerns raised in the review.
>
> ***
>
> > We don't need to annotate or learn a dense reward model, but sparse reward trajectories could solve all problems. The main weakness is that it is difficult to imagine how this result can be useful.
> > Could you provide any tangible procedure that could replace the PRM with ORM?
> > In practice, what will be the difficulty of applying this idea?
>
> Thanks for the feedback. We suspect that there might be some misunderstanding regarding the terminology and our focus. In this paper, we focus on statistical complexity by using an information-theoretic perspective to distinguish outcome/process supervision—that is—whether the external supervision signals are trajectory-wise or step-wise (see Section 2.2 for detailed explanation). Under our formulation, approaches that learn PRMs from outcome labels are considered outcome supervision, because these algorithmic interventions do not improve the statistical limits.
>
> Our primary focus is on statistical complexity rather than algorithmic design. The central takeaway is a theoretical result: RL from sparse trajectory rewards is not, information-theoretically, significantly harder than RL from dense step-level rewards, despite the ostensible gap in information.
>
> In terms of tangible procedures, many recent empirical studies that train PRMs using outcome labels [1,2,3] are, in our terminology, outcome-supervised (again, because the algorithmic aspects do not improve the statistical limits), which supports rather than contradicts our theoretical results. Our theory also aligns with the recent successes of outcome-based RL in the LLM industry (e.g., DeepSeek-R1, Kimi-K1.5), which further exemplify a tangible outcome-supervision procedure.
>
> ***
>
> > What assumption is not realistic?
>
>
> Note that our paper follows the most general assumptions of RL theory with general function approximation. In fact, perhaps surprisingly, our core technical contribution—the Change of Trajectory Measure Lemma (Lemma 3)—only requires the Markov property and does not rely on any structural assumption regarding the environment’s dynamics or function class. This means that Lemma 3 is widely applicable to any Markov decision process, beyond the problem in this paper.
>
> We suspect that this question from the reviewer arises because our key message (that outcome supervision is not much harder than process supervision) seems to contradict existing empirical results. However, this is just a matter of terminology, as we discussed above. Again, empirical successes in learning PRMs from outcome labels should be considered outcome supervision in our terminology, which supports rather than contradicts our theoretical results.
>
> ***
>
> References:
> [1] Luo, Liangchen, et al. "Improve mathematical reasoning in language models by automated process supervision." arXiv preprint arXiv:2406.06592 2 (2024).
> [2] Wang, Peiyi, et al. "Math-shepherd: Verify and reinforce llms step-by-step without human annotations." arXiv preprint arXiv:2312.08935 (2023).
> [3] Setlur, Amrith, et al. "Rewarding progress: Scaling automated process verifiers for llm reasoning." arXiv preprint arXiv:2410.08146 (2024).

---

### Official Review · Reviewer_F65b · 2025-03-14

**Overall Recommendation:** 2

**Summary:**

The paper compares process and outcome supervision in reinforcement learning from a theoretical perspective. The central result is showing that RL outcome rewards is no more statistically difficult than RL with step-level rewards, which is shown via a result that states that an offline RL dataset with outcome rewards can be transformed into a dataset with process rewards with minimal loss in statistical efficiency.

**Claims And Evidence:**

The theoretical claims all seem plausible, although I did not check the proofs in detail.

However, I think some of the claims in the introduction and general framing of the paper are too strong. In general, the results in the paper are about RL with goal-based reward and step-level reward. I don't think there is anything specific to RL with LLMs. Still the introduction heavily discusses work with LLMs suggesting particular applicability to that context. In fact, I think the applicability is probably lower because the reward signals in LLMs are often provided by human overseers.

The main claim of the paper is that we can draw conclusions about which of the two methods should work better in practice based on statistical difficult metrics of the learning problems. While I think the results are interesting and provide some evidence about this question, I think the introduction is vastly overclaiming the applicability to practical choices.

**Essential References Not Discussed:**

I was surprised to not see essentially any discussion of pre-LLM RL. In particular work on goal-conditioned RL and step-level feedback seems relevant. For example potential-based reward shaping is a way to transform a sparse ("outcome") reward into a shaped ("process") reward.

**Experimental Designs Or Analyses:**

No experiments.

**Methods And Evaluation Criteria:**

The proposed theoretical analysis is a valid way to gain insight about the comparison between process and outcome supervision.

However, the analysis neglects practically important aspects such as where do the process and outcome rewards come from. The paper shows a way of transforming process into outcome rewards and vice-versa but it is unclear what this implies in practice. For example, the quality of outcome and process rewards can vary widely if the rewards are given by humans. In some domains it is easier for humans to provide outcome rewards and in other domains it's easier to provide a process reward signal. So, overall I'm not drawing any strong conclusions from the results in this paper.

**Other Comments Or Suggestions:**

Overall I'm very unsure about this paper because I don't have the necessary expertise to evaluate the validity and importance of the theoretical claims. From a practical perspective I'm skeptical that the results are particularly useful. But I would support accepting the paper if other reviewers find the results interesting from a theoretical perspective.

**Other Strengths And Weaknesses:**

-

**Questions For Authors:**

-

**Relation To Broader Scientific Literature:**

The paper provides a theoretical perspective to compare outcome and process supervision which in this form is novel.

**Theoretical Claims:**

I did not check any of the proofs in detail and I do not have the relevant background to evaluate whether the proofs are correct.

---

> ### Author Rebuttal · Authors · 2025-04-01
>
> Thank you for the review. Below, we address the concerns raised in the review.
>
> ***
>
> > I think some of the claims in the introduction and general framing of the paper are too strong … I think the applicability is probably lower because the reward signals in LLMs are often provided by human overseers.
> > The analysis neglects practically important aspects such as where do the process and outcome rewards come from … overall I'm not drawing any strong conclusions from the results in this paper.
>
> If our understanding is correct, the main concern of the reviewer is that the outcome reward and process reward in practice often come from different sources with varying quality, so it is not practical to directly compare their sample complexity without considering these potential differences in quality.
>
> We appreciate this feedback; however, we would like to clarify that our paper primarily focuses on RL from outcome reward, because RL from process reward is already extensively studied in theory. This paper only considers the case where the outcome reward has the same quality as the process reward (i.e., outcome reward = sum of process reward), so that learning from process reward is always easier. We then investigate how much more difficult RL from outcome reward can be compared with process reward. This is well motivated by recent LLM reasoning problems, as checking the correctness of the entire solution is not more difficult than verifying each step.
>
> Our major result is both simple and somewhat counterintuitive: **we prove that RL from outcome reward is not much harder than RL from process reward, information-theoretically**, even though process reward seems to provide far more information for credit assignment. We believe this theoretical conclusion is significant, because it coincides with and explains recent surprising successes of outcome-based RL in the LLM industry (e.g., DeepSeek-R1, Kimi-K1.5).
>
> Finally, we emphasize that our submission is a theoretical paper, submitted to the theory category. Besides our main message, we present the novel Change of Trajectory Measure Lemma, which is the first result addressing trajectory-level change of measure with only step-level distribution shift. This lemma has broader applicability in RL theory beyond this paper. Hence, our theoretical results, rather than the outcome-to-process transformation itself, represent our core contribution and substantially deepen the theoretical understanding of reinforcement learning.
>
>
> ***
>
> > Pre-LLM RL discussion
>
> We do include substantial discussion of pre-LLM RL throughout the paper. For instance, Section 2 addresses coverage assumptions in the classic RL literature, and Section 3 discusses offline RL and preference-based RL. We also provide a Related Work section in the appendix, which includes references to pre-LLM RL research that is not covered in detail in the main text. Furthermore, we will add a discussion of RL with trajectory feedback and reward shaping to the Related Work section in our revision.

---

### Official Review · Reviewer_5vLh · 2025-03-17

**Overall Recommendation:** 3

**Summary:**

This paper challenges the conventional belief that process supervision (step-wise rewards) is statistically superior to outcome supervision (cumulative rewards) in reinforcement learning, particularly for complex tasks like LLM reasoning. The main contributions are:

1. **Theoretical Equivalence**: Under standard data coverage assumptions, outcome supervision is shown to have comparable statistical complexity to process supervision via a novel *Change of Trajectory Measure Lemma*. This lemma bounds trajectory-level distribution shifts using state-action concentrability, avoiding exponential dependence on horizon length.

2. **Advantage Functions as Optimal Process Rewards**: Theoretically, advantage functions of any policy can serve as optimal process reward models, while Q-functions may lead to sub-optimal results.

3. **Extensions to Preference-Based RL**: Improved sample complexity bounds for Direct Preference Optimization (DPO), replacing trajectory-level concentrability with state-action terms.

4. **Practical Algorithm**: A transformation method (Algorithm 1) that enables the use of process supervision algorithms with outcome supervision data.

## update after rebuttal
Maintain original score. See reasons in rebuttal comments.

**Claims And Evidence:**

The paper's claims are generally well-supported by theoretical analysis and proofs:

- **Supported Claims**:
  - The equivalence between outcome/process supervision under coverage (Theorems 1, 4) is rigorously proven using concentration inequalities and the Change of Trajectory Measure Lemma.
  - The optimality of advantage functions as reward models (Theorem 6) and the sub-optimality of Q-functions (Theorem 7) is validated via a constructed MDP counterexample.
  - The improved sample complexity for preference-based RL (Theorems 4, 5) is mathematically derived.

- **Unsupported Claims**:
  - Practical implications (e.g., reduced need for process annotations) remain speculative due to the lack of empirical validation.

The mathematical derivations are sound, though some proofs (particularly for Lemma 3) are quite technical and would benefit from more intuitive explanations.

**Essential References Not Discussed:**

The paper mentions but could more thoroughly discuss recent works on automated process supervision that generate step-wise rewards from outcomes, such as:

1. Wang et al. (2024) "Math-shepherd: Verify and reinforce LLMs step-by-step without human annotations"
2. Luo et al. (2024) "Improve mathematical reasoning in language models by automated process supervision"
3. Setlur et al. (2024) "Rewarding progress: Scaling automated process verifiers for LLM reasoning"

These works are briefly cited but not deeply analyzed, despite being highly relevant to the paper's claim that outcome supervision can be transformed into process supervision. A more detailed discussion would better contextualize the practical need for manual process labels and the empirical approaches already being used to address this issue.

**Experimental Designs Or Analyses:**

The paper is primarily theoretical and does not include empirical experiments. While the focus is theoretical, experiments on synthetic MDPs or LLM tasks would strengthen the practical relevance of the findings, especially given that they challenge conventional wisdom.

The absence of empirical validation is a limitation, particularly since the theoretical results suggest a different conclusion than what is often observed in practice, where process supervision frequently outperforms outcome supervision.

**Methods And Evaluation Criteria:**

The theoretical framework is appropriate, leveraging established concepts (state-action concentrability, offline RL theory) to analyze the statistical complexity of different supervision paradigms. The paper focuses on theoretical analysis rather than empirical evaluation, which is reasonable given the nature of the contributions.

The evaluation criteria, based on statistical complexity and sample efficiency, are appropriate for comparing the two supervision paradigms. However, the absence of experiments limits assessment of real-world applicability, especially given that the theoretical results contradict some empirical observations where process supervision often outperforms outcome supervision.

**Other Comments Or Suggestions:**

1. Section 4 on advantage function analysis could be better motivated and connected to the main thesis of the paper.

2. There are a few typos in the paper, including "contradistinction" (line 100) and "alterative" instead of "alternative" (line 171).

3. The paper would benefit from a discussion of limitations, particularly regarding the assumptions made in the theoretical analysis and how they might not hold in certain practical settings.

4. A simple illustrative example demonstrating the key insight of the paper would help readers grasp the main idea more intuitively.

**Other Strengths And Weaknesses:**

### Strengths:

1. **Novel Theoretical Insight**: The Change of Trajectory Measure Lemma is a significant technical contribution that challenges a widely held assumption about the statistical complexity of outcome vs. process supervision.

2. **Non-trivial Connections**: The paper establishes important connections between advantage functions and process rewards, providing theoretical justification for certain empirical approaches.

3. **Comprehensive Analysis**: The theoretical analysis covers various aspects of the problem, including extensions to preference-based learning and the role of advantage functions.

4. **Practical Algorithm**: Algorithm 1 provides a concrete way to transform outcome supervision data into process supervision data, making the theoretical insights actionable.

### Weaknesses:

1. **Heavy Reliance on Coverage Assumptions**: The results depend on state-action concentrability coefficients (Csa), which may not hold in practice, especially in complex environments like those encountered in LLM training.

2. **Lack of Empirical Validation**: The absence of experiments weakens the impact of the theoretical results, especially since they contradict some empirical observations.

3. **Limited Intuitive Explanations**: Some of the proofs, particularly for Lemma 3, are quite technical and would benefit from more intuitive explanations.

4. **Disconnect from Practice**: The paper could more explicitly address why, if outcome and process supervision are statistically equivalent, empirical results often show process supervision outperforming outcome supervision.

**Questions For Authors:**

1. How do your theoretical assumptions (e.g., Csa) align with real-world scenarios like LLM training, where state-action coverage is often poor? This would help clarify the practical applicability of your results.

2. Could the advantage function result (Section 4) be empirically validated on a simple MDP or LLM reasoning task? This would strengthen the connection between theory and practice.

3. Have you considered extending the analysis to settings with partial observability, which are common in sequential decision-making for LLMs? This would address a significant aspect of real-world applications.

Responses to these questions would clarify the applicability of the theoretical results and guide future empirical work in this area.

**Relation To Broader Scientific Literature:**

The paper effectively connects to several areas of reinforcement learning literature:

- **Offline RL**: Builds on work by Chen & Jiang (2019), Xie et al. (2021a), and others on concentrability coefficients and offline policy learning.

- **Off-Policy Evaluation**: Relates to work by Uehara et al. (2020) on the change of measure problem and the distinction between state-action and trajectory coverage.

- **Preference-Based RL**: Extends results to DPO (Rafailov et al., 2023) and provides improved sample complexity bounds.

- **Process vs. Outcome Supervision**: References empirical work showing the effectiveness of process supervision (Uesato et al., 2022; Lightman et al., 2023).

The paper's novelty lies in bridging outcome and process supervision via concentrability concepts and providing a theoretical foundation for understanding their relationship.

**Theoretical Claims:**

I checked the correctness of several key proofs:

1. **Change of Trajectory Measure Lemma (Lemma 3)**: The proof in Section B.1 is logically sound, using a careful decomposition of states into "good" and "bad" states and analyzing their probabilities under different policies. The technical approach effectively bounds trajectory-level distribution shifts using state-action concentrability.

2. **Theorem 1 (Learning a Reward Model from Total Reward)**: The proof correctly applies the Change of Trajectory Measure Lemma to bound the error between the true reward model and the learned reward model.

3. **Theorems 4 and 5 (Preference-Based RL)**: The proofs extend the main results to preference-based settings, correctly applying the Change of Trajectory Measure Lemma to derive improved sample complexity bounds.

4. **Theorem 6 (Advantage Function Learning)**: The proof correctly uses the performance difference lemma to show that the advantage function can serve as an optimal process reward model.

5. **Theorem 7 (Lower Bound on Failure of Using Q-Functions)**: The proof provides a valid counterexample showing that using Q-functions as reward models can lead to sub-optimal policies.

No apparent errors were detected in the proofs, though some sections could benefit from more intuitive explanations.

---

> ### Author Rebuttal · Authors · 2025-04-01
>
> Thank you for your review. Below, we address the concerns raised in the review.
>
> ***
>
> > more thoroughly discuss recent works on automated process supervision [1-3]
>
> Thanks for pointing these out, we will add more detailed discussion in the updated version.
>
> ***
>
> > Coverage Assumptions
>
> We would like to clarify that coverage assumptions are essential and widely used in the theory of offline RL (see discussion in Section 2.3). Our paper provides results on two kinds of coverage assumptions: in addition to $C_{\sf s,a}(\Pi,\pi_{\sf off})$ (coverage of all $\pi$ in $\Pi$) used in Section 3., we also have results in Appendix C that rely solely on $C_{\sf s,a}(\pi^*,\pi_{\sf off})$ (coverage for a single good policy $\pi^*$).
>
> We believe the latter notion should be considered practical even in the LLM regime, because it only requires the data distribution to cover a **single** good policy. For example, $\pi^*$ could be the best-of-N policy. In this case, $C_{\sf s,a}(\pi^*,\pi_{\sf off})$ is bounded by N.
>
> ***
>
> > Lack of Empirical Validation / Disconnect from Practice
>
> Thanks for pointing this out. We suspect that there might be some misunderstanding regarding the terminology. In this paper, we focus on statistical complexity by using an information-theoretic perspective to distinguish outcome/process supervision—that is—whether the external supervision signals are trajectory-wise or step-wise (see Section 2.2 for detailed explanation).
>
> Therefore, recent empirical results that learn PRMs from outcome labels [1,2] are actually outcome supervised in our language (because these algorithmic interventions do not benefit the statistical limits), which supports rather than contradicts our theoretical results. Our theory also coincides with the recent findings from DeepSeek-R1 and indicates that outcome annotation only induces a poly(H) additional cost of statistical complexity compared to step-by-step annotation (which can be easily bypassed if we have an “almost-free” rule-based outcome reward).
>
> ***
>
> > Limited Intuitive Explanations
>
> We agree the proof of Lemma 3 can be complicated and counterintuitive. We have re-written the proof sketch and will update it in our future version. Due to space constraints, we only include the most important intuition below:
>
> The key difficulty of implementing a trajectory-level change of measure is that small values of $|f(\tau)|$ do not necessarily ensure small values on every prefix $\tau_{1:h}$ or suffix $\tau_{h+1:H}$ (since cancellations could occur; for example, $a + b = 0$ does not imply $a = b = 0$). However, the Markov property ensures that if \(f\) has a small second moment under $\pi_{\sf off}$ and state $s_h$ is frequently visited, then $f$ cannot have high variance on either the prefix or the suffix through $s_h$. This is because high variance on any component would imply high variance for the entire trajectory.
>
> From that, we can apply the change of measure over these frequently encountered states from the offline policy to the objective policy, only paying the state-action concentrability. Afterward, we show that if all states along a trajectory satisfy the prefix/suffix trajectory low-variance property mentioned above, then the absolute value of the total rewards can be upper bounded as well.
>
> ***
>
> > about advantage function result (Section 4)
>
> There has been some work that explicitly validates the idea of using the advantage function in reward models [3], demonstrating an advantage over using the Q-function as a reward. However, it is still unclear which approach is theoretically correct, as using the Q-function as a reward also shows strong empirical potential [1,2]. In Section 4, our motivation is to clarify this A vs. Q debates: we formally prove that using the advantage function of any policy as a reward is correct in principle, whereas using the Q-function can fail in certain cases.
>
> ***
>
> > Decision making with partial observability
>
> Thank you for bringing this to our attention. This is an interesting research direction, and we plan to explore it in our future work.
>
> ***
>
> References:
> [1] Luo, Liangchen, et al. "Improve mathematical reasoning in language models by automated process supervision." arXiv preprint arXiv:2406.06592 2 (2024).
> [2] Wang, Peiyi, et al. "Math-shepherd: Verify and reinforce llms step-by-step without human annotations." arXiv preprint arXiv:2312.08935 (2023).
> [3] Setlur, Amrith, et al. "Rewarding progress: Scaling automated process verifiers for llm reasoning." arXiv preprint arXiv:2410.08146 (2024).

---

> > ### Comment · Reviewer_5vLh · 2025-04-02
> >
> > Thank you for your rebuttal addressing the concerns raised in my review. After considering your responses, I have the following thoughts:
> >
> > ## On Coverage Assumptions
> >
> > Your clarification about the two types of coverage assumptions is helpful. I appreciate the distinction between all-policy coverage (Csa(π, πoff) for all π) and single-policy coverage (Csa(π*, πoff)). The latter is indeed more practical, especially in LLM settings where coverage of a single good policy (like best-of-N) is more feasible. This addresses one of my main concerns about the practical applicability of your theoretical results.
> >
> > I still believe it would strengthen the paper to explicitly discuss how these coverage assumptions might manifest in practical LLM training scenarios, perhaps with concrete examples. This would help bridge the gap between theory and practice for readers less familiar with these concepts.
> >
> > ## On Empirical Validation and Terminology
> >
> > Thank you for clarifying the terminology. I now better understand that recent empirical approaches that learn PRMs from outcome labels [1,2] are considered "outcome supervised" in your framework, which indeed supports rather than contradicts your theoretical findings. This is an important distinction that could be made clearer in the paper to avoid similar misunderstandings by other readers.
> >
> > Your reference to DeepSeek-R1's findings about the poly(H) additional cost of statistical complexity for outcome annotation compared to step-by-step annotation is interesting and supports your theoretical results. Including this connection explicitly in the paper would strengthen the practical relevance of your work.
> >
> > ## On Intuitive Explanations
> >
> > I appreciate the more intuitive explanation of Lemma 3. Your explanation about how the Markov property ensures that frequently visited states cannot have high variance on either prefix or suffix trajectories helps clarify the counterintuitive nature of the result. This explanation would be valuable to include in the revised paper, as it makes the technical contribution more accessible.
> >
> > ## On Advantage Functions vs. Q-Functions
> >
> > Thank you for clarifying the motivation behind Section 4. Your explanation that this section aims to resolve the "A vs. Q debates" by formally proving that advantage functions are theoretically correct while Q-functions can fail in certain cases is compelling. This motivation could be made more explicit in the paper to better connect this section to the main thesis.
> >
> > The reference to empirical work [3] that validates the advantage function approach is helpful and strengthens the practical relevance of your theoretical findings.
> >
> > ## On Partial Observability
> >
> > I'm glad to hear you're interested in exploring decision-making with partial observability in future work. This would indeed be a valuable extension of your current results.
> >
> > ## Overall Assessment
> >
> > After considering your rebuttal, I maintain my recommendation of "Weak accept" but with increased confidence. Your clarifications have addressed my main concerns about:
> >
> > 1. The practical applicability of coverage assumptions
> > 2. The apparent disconnect between theory and empirical observations
> > 3. The motivation behind the advantage function analysis
> >
> > The paper makes a significant theoretical contribution by challenging conventional wisdom about process vs. outcome supervision, and your rebuttal has helped clarify how these theoretical insights connect to practical applications.
> >
> > For the revised version, I would recommend:
> > 1. Making the terminology distinctions clearer to avoid misunderstandings
> > 2. Including the more intuitive explanation of Lemma 3
> > 3. Explicitly connecting your theoretical results to recent empirical findings
> > 4. Better motivating Section 4 as resolving the "A vs. Q debates"
> > 5. More thoroughly discussing the recent works on automated process supervision
> >
> > These changes would strengthen the paper and make its contributions more accessible and impactful to the broader machine learning community.

---

> > > ### Author Response · Authors · 2025-04-06
> > >
> > > Thank you very much for your thoughtful follow-up comment and for acknowledging that our clarifications addressed your concerns. We also truly appreciate your detailed suggestions for improving the paper's accessibility, and we will incorporate them into our revision.
> > >
> > > Below is our complete version of the improved proof sketch for Lemma 3, and we hope it now provides a more intuitive explanation.
> > >
> > > ***
> > > > ### **Proof Sketch of Lemma 3 (New)**
> > >
> > > > **Insight I: Trajectory-level bound controls the second moment on prefixes and suffixes**
> > > > At first glance, small value $|f(\tau)|$ over the entire trajectory $\tau$ does *not* obviously guarantee that the value of $f$ on either (i) every prefix $\tau_{1:h}$ or (ii) every suffix $\tau_{h+1:H}$ is small. In principle, large positive and large negative portions of a single trajectory could “cancel” each other out, resulting in a small overall sum $|f(\tau)|=|f(\tau_{1:h})+f(\tau_{h+1:H})|$.
> > >
> > > > Crucially, however, thanks to the *Markov property*, we can argue that if $f$ has small second moment (under $\pi_{\sf off}$) and if a state $s_h$ is visited sufficiently often by $\pi_{\sf off}$, $f$ cannot have high variance on the prefix (leading up to $s_h$) and suffix (following $s_h$). Indeed, if the value of $f$ on the prefix (or suffix) has large variance, then conditioned on passing through $s_h$ that is visited sufficiently often by $\pi_{\sf off}$, the value of $f$ on the entire trajectory also has large variance, which directly implies the large variance (hence, large second moment) of $f(\tau)$. Hence, even though the trajectory-level bound looks coarse, it forces each state $s_h$ to have relatively stable partial sums in both the prefix and suffix directions under $\pi_{\sf off}$.
> > >
> > > > **Insight II: Layer-by-layer “locking” with only state-action coverage**
> > > > Next, we want to argue that if *all* states in a trajectory satisfy the above low-variance property (we call such states “good” states), then the reward of the entire trajectory cannot have large absolute value. We call this the “locking in” property here for brevity. In the following, we argue that “locking in” happens with high probability, even under policy $\pi$.
> > >
> > > > According to the earlier argument, “bad” states (opposite of “good” states) cannot have large visitation probability under $\pi_{\sf off}$. Then, by the definition of $C_{s,a}(\pi, \pi_{\sf off})$, which upper bounds the probability ratio between $\pi$ and $\pi_{\sf off}$ at any state, we conclude that such bad states also have low probability under $\pi$, up to a factor of $C_{s,a}(\pi,\pi_{\sf off})$. Thus, we avoid exponential blow-up over the horizon because we only “pay” for distribution shift at each individual $(s,a)$, rather than for entire trajectories:
> > > $\Pr_{\tau \sim \pi}(\text{bad}) \leq C_{s,a}(\pi,\pi_{\sf off})\cdot\Pr_{\tau \sim \pi_{\sf off}}(\text{bad})$.
> > >
> > > > Hence, with high probability, $\pi$ visits only “good” states throughout its trajectory, ensuring that each layer $h$ “locks in” a small partial sum (as both its prefix and suffix have low variance). (Footnote: Our formal proof also needs to consider the “good” state-action-state tuples, which are similar to “good” states but involve the $(s_h,a_h,s_{h+1})$ tuple. We omit the details here for brevity, and readers can refer to the full proof in Appendix B.1.)
> > > When we stitch these layers from $h = 1$ to $h = H$, the entire sum $f(\tau)$ is guaranteed to have small absolute values.
> > >
> > > ***
> > >
> > > On the other hand, we would still like to respectfully reiterate that our submission is fundamentally a theoretical work submitted to the Theory track. Although this paper is motivated by practical questions about different supervision paradigms for LLMs, we believe our core theoretical contributions, especially the Change of Trajectory Measure Lemma (Lemma 3), could stand on their own merit (even without any discussion or connection about empirical papers). This lemma challenges the conventional wisdom that learning from trajectory-wise feedback is intrinsically harder in RL theory (e.g., Section 4.3 of [4]). As Reviewer jQ9F also recognized in their positive evaluation ("lemma 3, is powerful and widely applicable even beyond offline RL... Future works will likely benefit from and build on these results"), the theoretical advancement itself is the major contribution of our paper, rather than providing insights for or explaining practical algorithms.
> > >
> > > Given the theoretical nature and scope of our paper, we would be very grateful if you could please reconsider your recommendation, especially based on our theoretical contributions like the Change of Trajectory Measure Lemma and its implications. We would be happy to discuss any further questions.
> > >
> > >
> > >
> > > ***
> > >
> > > Reference
> > >
> > > [4] Zhan, W., Uehara, M., Kallus, N., Lee, J. D., & Sun, W. Provable Offline Preference-Based Reinforcement Learning. In The Twelfth International Conference on Learning Representations.

---

### Decision · Program_Chairs · 2025-05-01

**Decision:**

Accept (poster)

**Comment:**

This work provides a comprehensive theoretical examination of process supervision in reinforcement learning. It establishes the statistical equivalence between outcome and process reward models, while demonstrating that process rewards approximating the advantage function is theoretically coherent. The theoretical insights presented in this work, in particular the Lemma 3, are highly valued by reviewers (5vLH, yWDQ, jQ9F). Despite some concerns in the practical algorithm design (Fb56, yWDQ), the paper's strength as a theoretical contribution with broad applicability (jQ9F) substantially outweighs these concerns, which is acknowledged by reviewers during the discussion period.

As a suggestion, this work may benefit from more discussion about relevant literature (5vLH). Besides [1-3], I also note [4] is related to the results in transforming outcome rewards for process rewards. Nonetheless, I think this work offers valuable insights that will benefit the broader research community, and recommend the acceptance of this work.

[1] Wang et al. Math-shepherd: Verify and reinforce LLMs step-by-step without human annotations. ACL 2024

[2] Luo et al. Improve mathematical reasoning in language models by automated process supervision. arXiv preprint arXiv:2406.06592 2 (2024).

[3] Setlur et al. Rewarding Progress: Scaling Automated Process Verifiers for LLM Reasoning. ICLR 2025.

[4] Feng et al. Step-by-Step Reasoning for Math Problems via Twisted Sequential Monte Carlo. ICLR 2025.